# Mesenchymal stromal cells alleviate depressive and anxiety-like behaviors via a lung vagal-to-brain axis in male mice

Jing Huang [1,2,5], Weijun Huang [1,2,5], Junzhe Yi [1,2,5], Yiwen Deng[3], Ruijie Li[1,2], Jieying Chen[1,2], Jiahao Shi[1,2], Yuan Qiu [1,2,4], Tao Wang [1,2,4], Xiaoyong Chen[1,2,4], Xiaoran Zhang [1,2,4] ✉ & Andy Peng Xiang [1,2,4] ✉

Major depressive disorder (MDD) is one of the most common and disabling mental disorders, and current strategies remain inadequate. Although mesenchymal stromal cells (MSCs) have shown beneficial effects in experimental models of depression, underlying mechanisms remain elusive. Here, using murine depression models, we demonstrated that MSCs could alleviate depressive and anxiety-like behaviors not due to a reduction in proinflammatory cytokines, but rather activation of dorsal raphe nucleus (DRN) 5-hydroxytryptamine (5-HT) neurons. Mechanistically, peripheral delivery of MSCs activated pulmonary innervating vagal sensory neurons, which projected to the nucleus tractus solitarius, inducing the release of 5-HT in DRN. Furthermore, MSC-secreted brain-derived neurotrophic factor activated lung sensory neurons through tropomyosin receptor kinase B (TrkB), and inhalation of a TrkB agonist also achieved significant therapeutic effects in male mice. This study reveals a role of peripheral MSCs in regulating central nervous system function and demonstrates a potential "lung vagal-to-brain axis" strategy for MDD.

Major depressive disorder (MDD) is a major public health issue[1]. Many people diagnosed with MDD are usually characterized by an enduring tendency to experience anxiety and to show poor resilience under stress[2]. Conventional antidepressants, selective serotonin reuptake inhibitors (SSRIs), have shown serious limitations, including delayed onset and no response in more than 30% of patients[3,4]. Ketamine and electroconvulsive therapy have shown rapid and robust antidepressant effects to reduce suicidality in patients, but potential addictive properties and the risk of inducing schizophrenia or seizures have raised concerns[5–7]. Therefore, scientists are still searching for stable, fast-acting antidepressant targets and compounds.

Several studies have indicated that mesenchymal stromal cells (MSCs) might exert beneficial effects due to their immunomodulatory and paracrine properties in rodent models of depression[8,9]. Specifically, a previous study demonstrated that implantation of encapsulated MSCs into the lateral ventricle enhanced neurogenesis by secretion of neurotrophic factors to counteract depressive-like behaviors in rats[10]. Additionally, one elegant study found that peripheral delivery of MSCs could improve depressive and anxiety-like behaviors in an inflammation-related depression model. The infused MSCs did not home to the brain but rather were phagocytosed by macrophages, which triggered an immunomodulatory cascade that resolved peripheral and central inflammation[11]. These findings indicated that

¹Center for Stem Cell Biology and Tissue Engineering, Key Laboratory for Stem Cells and Tissue Engineering, Ministry of Education, Sun Yat-Sen University, Guangzhou, Guangdong 510080, China. ²National-Local Joint Engineering Research Center for Stem Cells and Regenerative Medicine, Zhongshan School of Medicine, Sun Yat-Sen University, Guangzhou, Guangdong 510080, China. ³Key Laboratory of Medical Transformation of Jiujiang, Jiujiang University, Jiujiang, Jiangxi 332005, China. ⁴Department of Histoembryology and Cell Biology, Zhongshan School of Medicine, Sun Yat-Sen University, Guangzhou, Guangdong 510080, China. ⁵These authors contributed equally: Jing Huang, Weijun Huang, Junzhe Yi. ✉e-mail: zhangxr53@mail.sysu.edu.cn; xiangp@mail.sysu.edu.cn

intravenously administered MSCs might protect the distal brain from inflammation via indirect immunomodulatory mechanisms. However, only a subset of patients with MDD show hyperactivation of innate immunity[12]. The majority of patients with chronic anxiety and depression are in an immunosuppressive state that is characterized by susceptibility to infection[13–15]. Therefore, the specific effects and underlying mechanisms by which MSCs exhibit in stress-based models of depression remain largely unknown.

A recent report has shown that MSCs improve osteoarthritis pain by directly modulating joint-innervating sensory neurons, rather than by exerting anti-inflammatory or cartilage repair activities, which suggested the direct interaction between peripheral sensory neurons and MSCs via paracrine effects[16,17]. As most intravenously infused MSCs are distributed in the lungs[18], we hypothesized that MSCs may interact with abundant lung-innervating sensory neurons and further transmit a signal to the central nervous system. At present, little is known about how sensory innervation of the lung contributes to regulating reward circuits, and it is unclear whether the pulmonary sensory innervation contributes to the ability of MSCs to regulate behavioral patterns in murine depression models.

Here, using chronic restraint stress (CRS) and repeated social defeat (RSD) murine models of depression, we study the antidepressant and anxiolytic effects of MSCs and the underlying mechanisms. We find peripheral delivery of MSCs activates pulmonary innervating vagal sensory neurons, which project to the nucleus tractus solitarius, further inducing the release of 5-hydroxytryptamine (5-HT) in the dorsal raphe nucleus (DRN). In addition, MSCs activate lung-innervating vagal sensory neurons partially through the BDNF-TrkB signaling pathway. Inhalation of a TrkB agonist improves the aberrant behavioral patterns of CRS mice. Overall, this study highlights the role of peripheral MSCs in regulating central nervous system function and the potential "lung vagal-to-brain axis" therapeutic strategy in depression models.

## Results

### Intravenous administration of MSCs attenuates depressive and anxiety-like behaviors in mouse models

To examine the antidepressant and anxiolytic effects of MSCs, a total of $1 \times 10^6$ human dermal fibroblasts (HDFs) or bone marrow-derived MSCs were intravenously administered on day 11 of chronic restraint stress (CRS) in model mice, as outlined in Fig. 1a. Compared with the CRS group, MSCs, but not HDFs, significantly alleviated the weight loss seen alongside depression in this model (Fig. 1b). In the open field test (OFT), the total distance traveled by the mice was comparable across the four groups. CRS mice showed little interest in entering the center field of the chamber, while those infused with MSCs exhibited marked increases in the time spent and distance traveled in the center field (Fig. 1c, d). Similar to the results of the OFT, mice exposed to CRS spent less time in the open arms of the elevated plus maze (EPM) than mice in the control group, whereas MSC treatment, but not HDF injection, notably increased the time spent in the open arms (Fig. 1e, f). The OFT and EPM results indicated that MSCs might exert anxiolytic effects in CRS mice. Furthermore, infusion of MSCs, but not HDFs, into CRS mice significantly decreased the immobility time in the tail-suspension test (TST), which suggested that MSCs might exert antidepressant effects in CRS mice (Fig. 1g).

We used the repeated social defeat (RSD) mouse model, which shows decreased sociability and increased anxiety behaviors[19] (Supplementary Fig. 1a), to detect the ability of MSCs to regulate social avoidance and anxiety-like behaviors in mice. In the social avoidance test, RSD mice spent less time close to an unfamiliar CD1 mouse within a designated interaction zone, as illustrated in Supplementary Fig. 1b. After MSC administration, the mice showed a prominently increased willingness to interact with the CD1 mice (Supplementary Fig. 1b, c). Moreover, RSD mice spent less time in open arms in the EPM test, and these anxiety-like behaviors were decreased by MSC infusion

(Supplementary Fig. 1d, e). Unstressed mice with MSC or HDF infusion showed a similar weight change trend and behavior pattern in the EPM (Supplementary Fig. 1f, g). Overall, these results suggest that MSCs, but not HDFs, alleviate the depressive and anxiety-like behaviors in CRS and RSD mice.

### 5-HT neurons in the dorsal raphe nucleus (DRN) are activated after MSC treatment

MSCs have been reported to improve the aberrant behavioral patterns in a mouse model of depression by modulating inflammatory processes[20,21]. We next examined whether the alleviation of murine depressive and anxiety-like behaviors in MSC-treated stressed mice resulted from anti-inflammatory effects. After 4 h of restraint stress, we detected IL-6 and TNF-α expression levels and found that IL-6 was significantly increased in serum and brain homogenate supernatant, which could be alleviated by MSC injection (Supplementary Fig. 2a, b). In contrast, MSC injection barely affected the level of TNF-α at 4 h of restraint stress (Supplementary Fig. 2c, d). Our data also showed that acute stress significantly activated microglia, and MSC injection alleviated neuroinflammation. However, 11 days of restraint stress did not induce significant microglial activation (Supplementary Fig. 2e, f). The data indicated that the inflammation level of mice depends on the stress duration, and only at the early stage of stress, MSCs showed a notable anti-inflammatory effect. Similarly, we observed that after 14 days of CRS or 7 days of RSD, IL-6, and TNF-α in the stress-treated mice were not significantly different from those in the normal and cell therapy groups (Supplementary Fig. 2g–j). Taken together, these findings suggest that other mechanisms might explain the beneficial effects of MSCs in chronic stress-induced depressive mice.

Stress-induced depressive and anxiety-like behaviors are commonly associated with the dopaminergic system[22], hypothalamic-pituitary-adrenal (HPA) axis[23,24], and serotonergic system[25,26], so we hypothesized that MSCs could exert an antidepressant and anxiolytic function by modulating regions relevant to these systems. Our results demonstrated that MSCs had little influence on the c-Fos expression (a marker for neuronal activation) of dopamine neurons in the ventral tegmental area (VTA) or the concentration of dopamine in brain homogenates (Supplementary Fig. 3a–c). We also tested whether restraint stress could activate corticotrophin-releasing hormone (CRH) neurons in the paraventricular nucleus (PVN) but found that MSCs failed to decrease c-Fos expression in CRH neurons or the CRH concentration in brain homogenates (Supplementary Fig. 3d–f). However, after MSC injection, the dorsal raphe nucleus (DRN) region of the serotonergic system exhibited obvious neural activation (Fig. 1h–j) and a time-dependent increase in c-Fos expression (Supplementary Fig. 3g, h). The neuronal activation observed in the DRN was predominantly located beneath the midbrain aqueduct, which mainly contains 5-hydroxytryptamine (5-HT) neurons[27]. As expected, a remarkable increase in the colocalization of c-Fos and 5-HT was observed after MSC injection (Fig. 1k, l, and Supplementary Fig. 3i). To further confirm that 5-HT neurons were activated after MSC administration, we performed in vivo electrophysiological recordings in the DRN. The results showed that the firing rate of these neurons was increased upon MSC infusion (Fig. 1m, n, and Supplementary Fig. 4a–c). Moreover, we also performed patch-clamp recording to explore the intrinsic excitability of DRN 5-HT neurons in the control, CRS, and CRS + MSC groups. Our data showed that CRS induced a decrease in the firing frequency of 5-HT neurons, and MSC infusion could enhance the firing frequency of DRN serotonergic neurons (Supplementary Fig. 4d–f). Given our electrophysiological results showing that DRN neurons are activated upon MSC administration and in situ c-Fos immunohistochemistry data showing that these activated signals are mainly from 5-HT⁺ neurons, we next assessed the real-time activity of 5-HT^DRN neurons using fiber photometry (Fig. 1o).

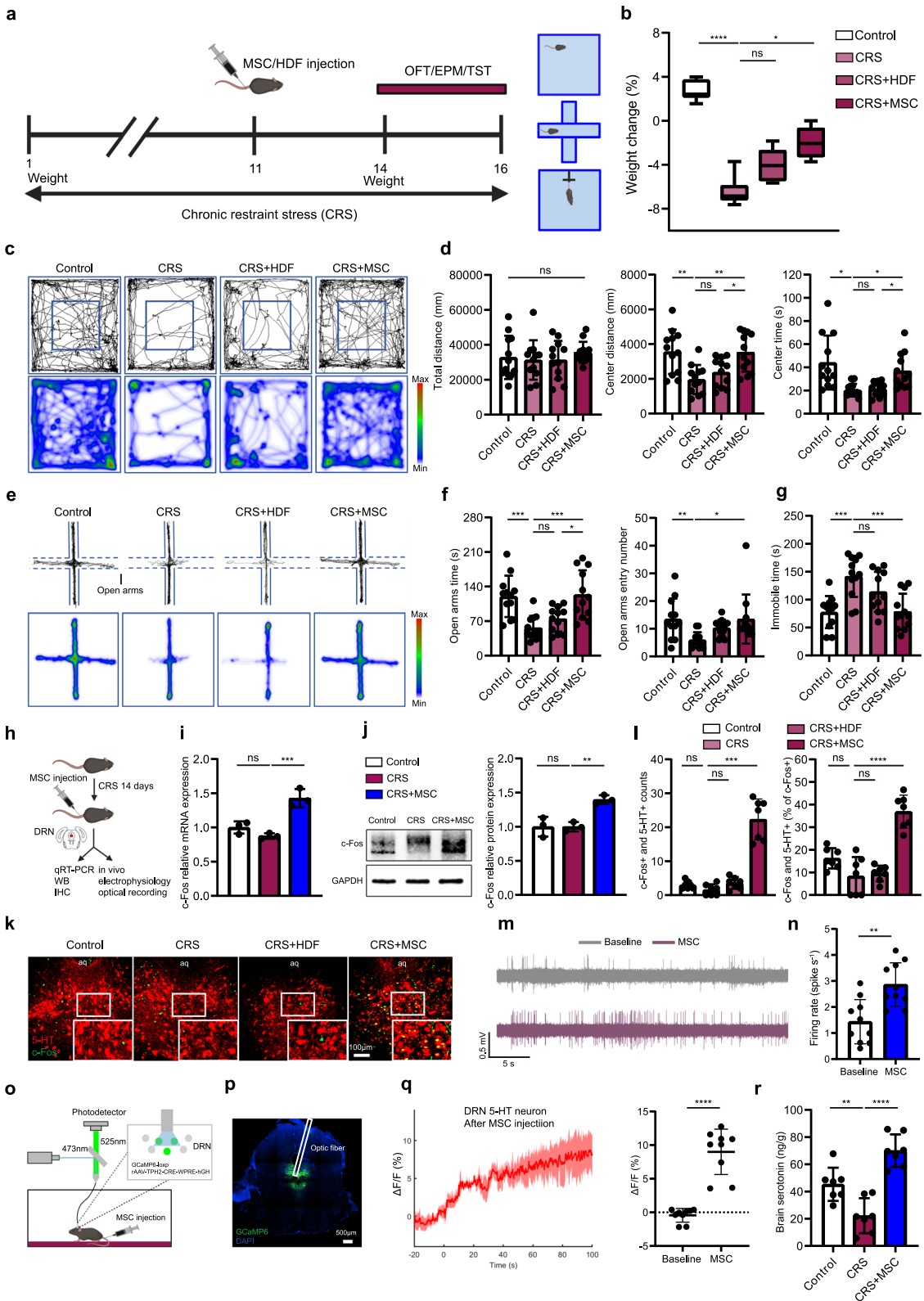

Consistent with the above findings, 5-HT[DRN] neurons were markedly activated upon MSC infusion (Supplementary Fig. 4g, and Fig. 1p, q). Of note, CRS mice tended to exhibit a lower brain level of 5-HT, but this level increased at least 3-fold upon MSC administration (Fig. 1r, and Supplementary Fig. 4h). Overall, these data indicate that serotonergic neurons within the DRN are activated by intravenous MSC infusion.

## MSCs potentially exert antidepressant and anxiolytic effects through 5-HT[DRN] neurons in CRS mice

To investigate the potential role of 5-HT[DRN] neurons in the antidepressant and anxiolytic effects of MSCs, we first constructed a serotonergic neuron-lesioned mouse model by performing stereotaxic injection of the selective serotonergic neurotoxin, 5, 7-dihydroxytryptamine (5, 7-DHT), into the DRN. One week after post-

**Fig. 1 | MSCs attenuate depressive and anxiety-like behaviors. a** Schematic of the experimental timeline. HDF, human dermal fibroblast. **b** Body weight changes after CRS. Kruskal-Wallis test: $^{****}P = 0.0000094$. $n = 8$ mice. **c, d** Representative traces and statistical analysis of OFT. Total distance, one-way ANOVA: $F_{(3, 44)} = 0.46$ ns $P = 0.71$. Center distance, one-way ANOVA: $F_{(3, 44)} = 7.4$ $^{***}P = 0.00039$. Center time, Brown-Forsythe ANOVA test: $^{***}P = 0.00094$. $n = 12$ mice. **e, f** Representative traces and statistical analysis of EPM. Open arms time, one-way ANOVA: $F_{(3, 44)} = 9.9$ $^{****}$ $P = 0.000042$. Open arms entry number, Kruskal-Wallis test: $^{**}P = 0.0018$. $n = 12$ mice. **g** Statistical analysis of TST. One-way ANOVA: $F_{(3, 44)} = 10.16$ $^{****}P = 0.000033$. $n = 12$ mice. **h** Schematic illustrating experimental protocol. **i** qRT-PCR detection. One-way ANOVA: $F_{(2, 6)} = 27.49$ $^{***}P = 0.0010$. $n = 3$ biologically independent samples. **j** Western blot detection. One-way ANOVA: $F_{(2, 6)} = 15.94$ $^{**}P = 0.0040$. $n = 3$

biologically independent samples. **k, l** Representative images and quantification of 5-HT and c-Fos colocalization. Scale bar, 100 μm. c-Fos+ and 5-HT+ count, Brown-Forsythe ANOVA test: $^{****}P = 0.0000044$. c-Fos+ and 5-HT+ (% of c-Fos +), one-way ANOVA: $F_{(3, 24)} = 31.38$ $^{****}P = 0.000000018$. $n = 7$ mice. **m** Representative traces of electrical activity in stressed mice. **n** Firing rates. Two-tailed $t$ test: $t = 3.785$, df $= 18$, $^{**}$ $P = 0.0014$. $n = 10$ neurons from three independent experiments. **o** Schematic diagram for optic fiber recordings. **p** Representative image showing GCaMP6 expression in DRN. **q** Representative traces of integrated calcium signals from 5-HT$^{DRN}$ neurons. Statistical analysis. Two-tailed $t$ test: $t = 8.036$, df $= 16$, $^{****}P = 0.00000052$. $n = 9$ mice. **r** The 5-HT level change. One-way ANOVA: $F_{(2, 18)} = 26.29$ $^{****}$ $P = 0.0000045$. $n = 7$ mice. Illustrations created with BioRender.com. Source data are provided as a Source Data file.

surgery, the mice received CRS as outlined in Fig. 2a. We also performed a two-virus injection to selectively ablate 5-HT$^{DRN}$ neurons (rAAV-EF1a-DIO-taCasp3 and rAAV-TPH2-CRE) (Fig. 2a). Both methods could induce a significant elimination of 5-HT$^{DRN}$ neurons (Fig. 2b, c and Supplementary Fig. 5a). The 5,7-DHT lesion protocol did not cause evident anxiety-like behaviors, as measured by the distance traveled and time spent in the center field of the OFT. However, MSC injection did not effectively increase the distance traveled or time spent in the center field for CRS mice injected with 5,7-DHT (Fig. 2d, e). Similarly, MSCs did not suppress CRS-induced anxiety-like behaviors in mice subjected to 5-HT$^{DRN}$ neuron ablation with AAVs, as evaluated by the distance traveled and time spent in the center field (Fig. 2f, g). Under the 5,7-DHT-induced absence of 5-HT$^{DRN}$ neurons, we observed a similar trend in anxiety-like behaviors, as tested by time spent in and entry numbers to open arms of the EPM (Fig. 2h, i). Consistent with the results obtained from the OFT and EPM, infusion of MSCs failed to relieve CRS-induced depressive-like behavior in the TST in mice with DRN serotonergic neuron lesions induced via 5,7-DHT (Fig. 2j). We also failed to observe any marked difference between the AAV + CRS and AAV + CRS + MSC groups in open arms time and entry numbers (Fig. 2k, l). Infusion of MSCs also failed to rescue depressive-like behaviors in the TST in CRS mice with AAV-induced 5-HT$^{DRN}$ ablation (Fig. 2m). In addition, to further determine whether the activation of 5-HT neurons induced by MSCs is necessary for the MSC-mediated antidepressant and anxiolytic effects, we performed hM4Di-mediated 5-HT+ neuron silencing in the presence of MSC infusion (Supplementary Fig. 5b, c). We found that MSC did not improve CRS-induced depressive and anxiety-like behaviors in CRS mice with hM4Di silencing of 5-HT+ neurons, as shown in the OFT, EPM, and TST (Supplementary Fig. 5d–h). In conclusion, MSCs potentially exert antidepressant and anxiolytic effects through 5-HT$^{DRN}$ neurons in the CRS model.

## Lung-innervating sensory neurons mediate the activation of 5-HT$^{DRN}$ neurons by peripheral MSCs

To determine whether the activation of 5-HT$^{DRN}$ neurons was the result of active MSCs homing toward the brain, biodistribution studies were undertaken following infusion. GFP-labeled MSCs were delivered intravenously to the stressed mice, and histologic sections were analysed to identify primary sites of MSC engraftment. MSCs were distributed primarily among peripheral tissues, especially in the lung, and were not detected in the brain (Fig. 3a, b). This biodistribution pattern suggests that the behavioral improvement and 5-HT$^{DRN}$ neuron activation triggered by intravenous MSC administration did not arise due to active homing of infused cells to the brain.

Lungs are richly innervated with a range of sensory nerve fibers that convey peripheral information to the central nervous system[28]. Therefore, we questioned whether sensory neurons might provide a bridge between lung-distributed MSCs and 5-HT$^{DRN}$ neurons. To address this question, we used a secreted fluorescent mCherry protein containing a modified lipid-permeable transactivator of transcription (TATk) peptide (sLP-mCherry) to label GFP-MSCs, and detected its signals to directly identify their neighboring cells in vivo (Fig. 3c).

Indeed, intravenous injection of labeling-GFP MSCs (GFP+mCherry+) released sLP−mCherry and efficiently labeled surrounding host tissue cells (Fig. 3d). We further found that the GFP-mCherry+ cells colocalized with the panneuron marker, beta-III tubulin (Fig. 3e), suggesting that components secreted by MSCs might be taken up by pulmonary nerve fibers. The afferent activities arising from the lung are conducted primarily by branches of vagus nerves[29]. We hypothesized that MSCs may influence pulmonary vagal sensory nerves. Indeed, we observed that MSCs could be located close to pulmonary VGLUT2+ (a marker for vagal sensory neurons) innervations (Fig. 3f). To further determine whether MSCs directly elicited responsiveness in sensory fibers, we crossed *VGLUT2-IRES-Cre* mice with *GCaMP6-loxp* mice, and recorded the activity of lung sensory fibers in ex vivo lung slices from *VGLUT2-GcaMP6* mice (Supplementary Fig. 6a–c). We found that vagal sensory fibers were stimulated by MSCs, as measured by GCaMP6 fluorescence intensity (Fig. 3g and Supplementary Fig. 6d). To further confirm the positive response of vagal sensory neurons to MSC injection, we observed a notable increase in c-Fos expression in the nodose ganglia, where the cell bodies of vagal sensory neurons are located (Fig. 3h).

As vagal afferents primarily project to the nucleus tractus solitarius (NTS) in the medulla[30], we detected c-Fos expression in the NTS after MSC administration (Fig. 3i, l), and also performed vagotomy (left or right vagal nerves) before MSC injection (Fig. 3j). Our data showed that intravenous MSC application activated neurons in the NTS, and that unilateral vagotomy abolished the activation of NTS neurons on the same side (Fig. 3i, k, l) and ultimately reduced c-Fos signals and the colocalization of 5-HT and c-Fos in the DRN (Fig. 3m−o). In summary, these results indicate that MSCs in the lung positively activate pulmonary afferent neurons and may thereby convey information to the central nervous system (CNS).

## MSCs relay neural signals to the DRN via pulmonary sensory nerves

To investigate how pulmonary sensory afferents relay peripheral signals to 5-HT$^{DRN}$ neurons, we used a retrograde transsynaptic viral tracer to map the neural circuit. We injected enhanced green fluorescent protein-expressing recombinant pseudorabies virus (PRV) Bartha strain into the DRN region. Our results revealed that infection of mice with PRV-EGFP yielded bright GFP fluorescence in fibers and cells of the DRN and that these were mainly 5-HT neurons (almost 80%) (Fig. 4a). At 96 h post-infection, the PRV had descended from the DRN into the medial vestibular nucleus, nucleus prepositus, NTS, nodose ganglia, and lung (Supplementary Fig. 7a and Fig. 4b−f). Moreover, RFP-overexpressing MSCs (RFP-MSCs) were located close to the PRV-EGFP-marked fibers in the lungs (Fig. 4g). To confirm this circuit, we injected anterogradely transported and polysynaptic herpes simplex viruses (HSV-EGFP) into the NTS (Fig. 4h, i). At 72 h post-injection, the infection extended to DRN targets, and several infected neurons in the DRN were found to be serotonergic (Fig. 4j). Collectively, these findings allowed us to map the neural circuit from lung afferents to the DRN by injecting PRV-EGFP and HSV-EGFP, which enabled retrograde and anterograde tracing of neural circuits respectively. Our results

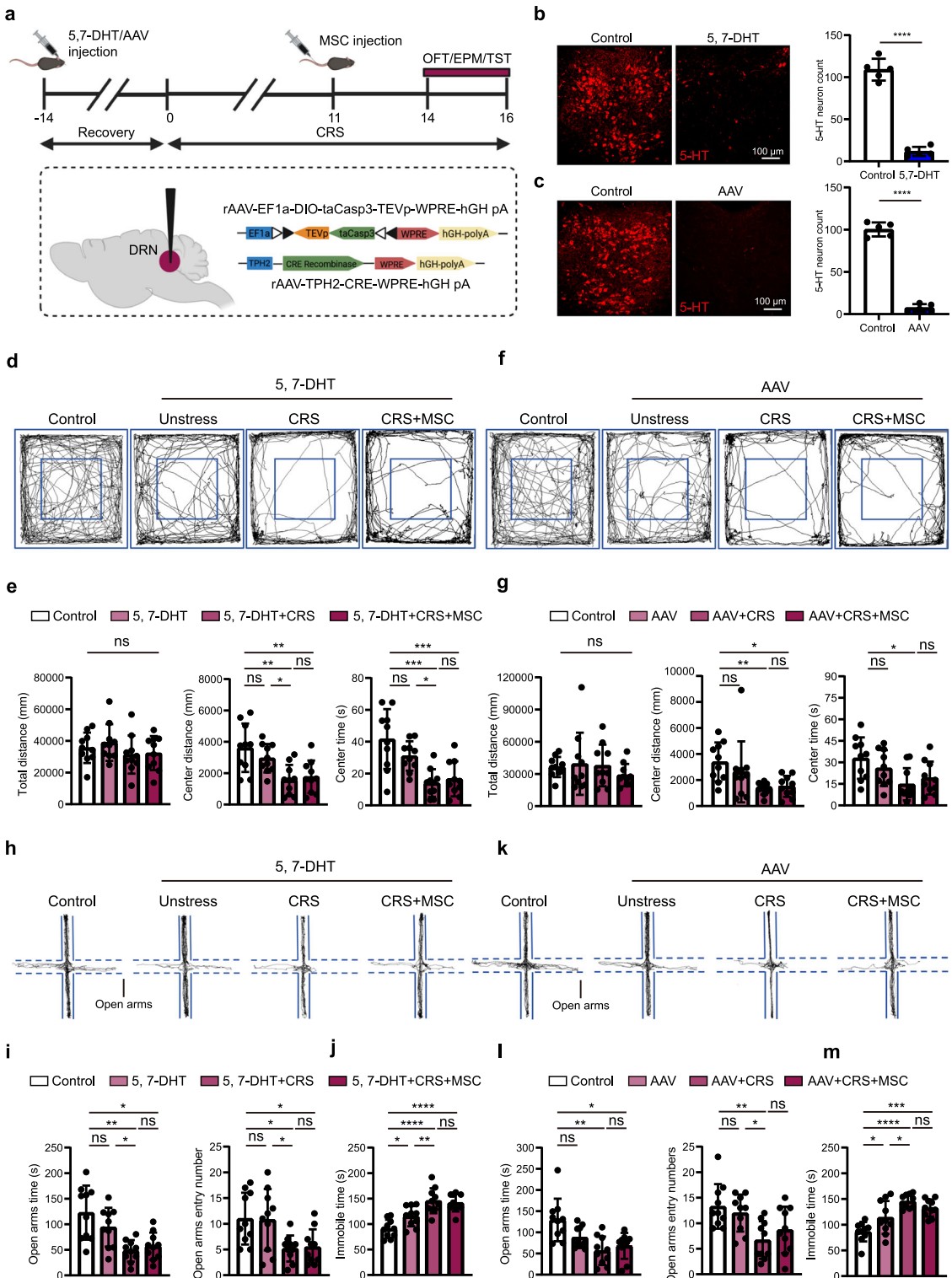

**Fig. 2 | MSCs potentially exert an antidepressant effect through 5-HT<sup>DRN</sup> neurons. a** Schematic diagram showing the experimental procedures. **b** Representative images and quantification showing that 5-HT neurons in the DRN were eliminated by 5, 7-DHT injection. Scale bar, 100 μm. Two-tailed $t$ test: $t = 15.43$, df = 8, ****$P = 0.00000031$. $n = 5$ mice. 5, 7-DHT, 5,7-Dihydroxytryptamine. **c** Representative images and quantification showing that 5-HT neurons in the DRN were eliminated by AAV injection to overexpress Caspase3. Scale bar, 100 μm. Two-tailed $t$ test: $t = 22.12$, df = 8, ****$P = 0.000000018$. $n = 5$ mice. **d**, **f** Representative activity tracking in the OFT. **e**, **g** Statistical analysis of the total distance, the center distance, and the center time. **e** Total distance, one-way ANOVA: $F_{(3, 36)} = 0.9807$ ns $P = 0.41$. **e** Center distance, one-way ANOVA: $F_{(3, 36)} = 7.363$ ***$P = 0.00057$. **e** Center time, one-way ANOVA: $F_{(3, 36)} = 10.50$ ****$P = 0.000042$. **g** Total distance, Kruskal-Wallis test: ns

$P = 0.46$. **g** Center distance, Kruskal-Wallis test: **$P = 0.0043$. **g** Center time, one-way ANOVA: Treatment $F_{(3, 36)} = 4.097$ *$P = 0.013$. $n = 10$ mice. **h**, **k** Representative activity tracking in the EPM. **i**, **l** Statistical analysis of the time spent in and entry numbers to open arms. (i) Open arms time, Brown-Forsythe ANOVA test: ***$P = 0.00037$. (i) Open arms entry number, one-way ANOVA: $F_{(3, 36)} = 5.219$ **$P = 0.0043$. **l** Open arms time, Kruskal-Wallis test: **$P = 0.0030$. **l** Open arms entry number, one-way ANOVA: $F_{(3, 36)} = 5.485$ **$P = 0.0033$. $n = 10$ mice. **j**, **m** Statistical analysis of the immobile time in the TST. **j** Immobile time, one-way ANOVA: $F_{(3, 36)} = 16.4$ ****$P = 0.00000070$. **m** Immobile time, one-way ANOVA: $F_{(3, 36)} = 14.25$ ****$P = 0.0000028$. $n = 10$ mice. Illustrations created with BioRender.com. Source data are provided as a Source Data file.

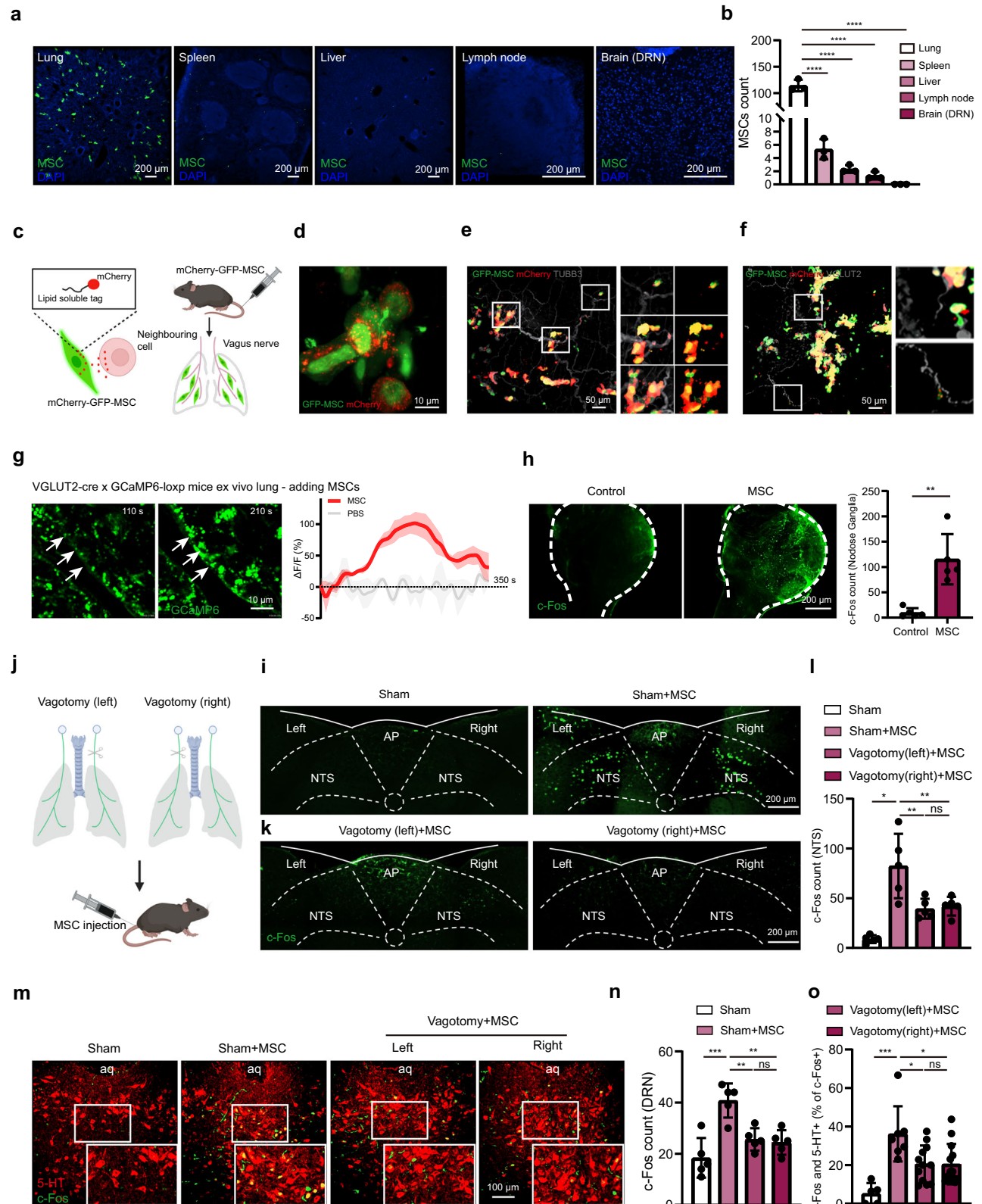

confirm that MSC-triggered neural signaling may run from afferent nerves innervating the lungs to 5-HT[DRN] neurons in the brain.

## MSCs activate 5-HT neurons via brain-derived neurotrophic factor (BDNF)

To investigate the in vivo biological characteristics of MSCs, we performed RNA-seq experiments to compare the transcriptional profiles between cultured GFP+ MSCs and GFP+ MSCs isolated from the mouse lungs 3 days after transplantation. We found that the two groups of MSCs had similar expression levels of the MSC markers CD29, CD44, and CD90 and were negative for CD3, CD11b, CD19, and CD34 (Supplementary Fig. 8a). In addition, neither group of cells expressed the osteogenic, chondrogenic, or adipogenic related genes (Supplementary Fig. 8b), indicating that there was no aberrant differentiation of

**Fig. 3 | Lung-innervating sensory neurons mediate the activation of 5-HT$^{DRN}$ neurons by peripheral MSCs. a** Representative images from three independent experiments showing GFP-MSC distribution. Scale bar, 200 μm. **b** Quantification of GFP-MSCs. One-way ANOVA: F$_{(4, 10)}$ = 303.6 *** $P$ = 0.00000000022. $n$ = 3 mice. **c** Label design for mCherry-GFP-MSCs. **d** Representative image from three independent experiments showing that GFP-MSCs co-expressed mCherry. Scale bar, 10 μm. **e** Representative image from three independent experiments showing the colocalization between mCherry secreted from MSCs and pulmonary nerves (TUBB3). Scale bar, 50 μm. **f** Representative image showing from three independent experiments the colocalization between mCherry secreted from MSCs and pulmonary vagal nerves (VGLUT2). Scale bar, 50 μm. **g** Representative sequential images from three independent experiments displaying calcium signal response to MSC suspensions inside the ex-vivo lungs of VGLUT2-GCaMP6 mice. Arrow points to a region showing an increase in GCaMP6 fluorescence. Scale bar, 10 μm. $n$ = 3 mice. Right, real-time changes in fluorescence intensity were expressed as percentage changes over baseline ((F-F0)/F0). **h** Representative images and statistical analysis showing the expression of c-Fos in nodose ganglia. Scale bar, 200 μm. Two-tailed $t$ test: $t$ = 4.685, df = 8 ** $P$ = 0.0016. $n$ = 5 mice. **i** Representative images from three independent experiments showing the NTS c-Fos expression. Scale bar, 200 μm. AP, area postrema. **j** The experimental procedures. **k** Representative images from three independent experiments showing the NTS c-Fos expression. Scale bar, 200 μm. **l** Statistical analysis of the NTS c-Fos expression. Brown-Forsythe ANOVA test: ** $P$ = 0.0046. $n$ = 5 mice from three independent experiments. **m** Representative images showing the colocalization of 5-HT and c-Fos in the DRN. Scale bar, 100 μm. aq, mesencephalic aqueduct. **n** Statistical analysis of the DRN c-Fos expression. One-way ANOVA: F$_{(3, 16)}$ = 12.32 *** $P$ = 0.00020. $n$ = 5 mice. **o** Statistical analysis of the colocalization of 5-HT and c-Fos. One-way ANOVA: F$_{(3, 33)}$ = 8.545 *** $P$ = 0.00024. $n$ = 5, 7, 12, 13 mice separately. Illustrations created with BioRender.com. Source data are provided as a Source Data file.

the transplanted MSCs. Compared with the control group, the transplanted cells showed a similar gene expression pattern of survival-related genes[31] and secreted protein genes (Supplementary Fig. 8c, d). Both of these results indicate that the transplanted GFP-MSCs still retain MSC properties.

To discover the potential candidates participating in the MSC-mediated lung afferent neuron activation, we analysed the transcriptional profiles of MSCs. According to the Ingenuity Pathway Analysis (IPA) database (licence ID: 46874), the Venn diagram indicated that MSC-secreted protein, brain-derived neurotrophic factor (BDNF), was associated with major depressive and anxiety disorders (Fig. 5a). Considering that BDNF is widely accepted for its involvement in resilience and antidepressant drug action[32], we chose BDNF as the candidate target for further study. In addition, we observed the colocalization of BDNF and the GFP signal of MSCs in lung sections by immunofluorescence staining (Supplementary Fig. 9a). We then sorted the GFP-positive MSCs from mouse lungs using FACS 3 days after the transplantation and confirmed the BDNF expression at the protein and mRNA levels (Supplementary Fig. 9b, c). Both of these results indicate that the transplanted MSCs are still alive and functional. More importantly, the presence of MSCs considerably increased the phosphorylation levels of the BDNF receptor, tropomyosin receptor kinase B (TrkB) in the VGLUT2 nerves compared to the control group (Fig. 5b, c). These findings indicate that peripheral MSCs may activate lung-innervating vagal sensory neurons via BDNF-TrkB signaling.

To assess the role of BDNF in the MSC-mediated regulation of the 5-HT$^{DRN}$ neurons, we knocked down the expression of BDNF in MSCs (MSC$^{BDNFsi1}$, MSC$^{BDNFsi2}$) by using two different short-interfering RNA (siRNA) sequences. Compared to MSCs transfected with scramble siRNAs (MSC$^{NC}$ negative control), MSC$^{BDNFsi1}$ and MSC$^{BDNFsi2}$ expressed lower levels of BDNF (Fig. 5d–g). Moreover, MSC$^{BDNFsi1}$ and MSC$^{BDNFsi2}$ showed little ability to activate 5-HT$^{DRN}$ neurons (Fig. 5h, i). When we investigated the importance of the BDNF-TrkB axis in the ability of MSCs to improve depressive and anxiety-like behaviors, we found that BDNF-knockdown MSCs could not rescue the CRS-induced decreases in distance traveled and time spent in the center field of the OFT, time spent in and entry numbers to open arms in the EPM, and the increase in immobile time in the TST (Supplementary Fig. 10). To further confirm the results, we used ANA12, which is a small-molecule antagonist of TrkB. The delivery of ANA12 by nebulizer inhalation before MSC infusion was found to abrogate MSC-mediated 5-HT neuronal activation (Fig. 5d, j, k). These data indicate that the BDNF-TrkB axis of the lungs is essential for MSC-mediated 5-HT neuronal activation, antidepressant and anxiolytic effects.

### Inhalation of a TrkB agonist, 7, 8-DHF, alleviates depressive and anxiety-like behaviors in CRS mice

Given our data suggesting that MSCs mediate 5-HT$^{DRN}$ neuronal activation and behavioral improvements via the BDNF-TrkB axis of lung afferents, we hypothesized that inhalation of the TrkB agonist, 7, 8-DHF, could improve depressive and anxiety-like behaviors and thus represent a promising treatment for depression. To explore this possibility, CRS-induced depression model mice were generated by 14 consecutive days of restraint stress and a nebulizer was used to administer 10 mg of 7, 8-DHF per kg of body weight, or an equal volume of vehicle, via inhalation (Fig. 6a). First, we detected the activation of the 5-HT$^{DRN}$ and changes in brain 5-HT concentrations. Our data showed that 7, 8-DHF inhalation treatment could activate 5-HT neurons in the DRN and improve the brain level of 5-HT (Fig. 6b, c). Next, using the OFT, EPM, TST, and forced-swimming test (FST), we assessed the therapeutic effects of 7, 8-DHF inhalation on day 15 (Fig. 6a). In the OFT, the inhalation of 7, 8-DHF significantly suppressed CRS-induced anxiety-like behaviors, as measured by the distance traveled and time spent in the center field of the chamber (Fig. 6d, f). Inhalation of the TrkB agonist also increased the preference of CRS mice for the open arms of the EPM, suggesting that the anxiety-like behaviors caused by chronic restraint stress were relieved by 7, 8-DHF inhalation treatment (Fig. 6e, g). Inhalation of nebulized 7, 8-DHF was also sufficient to rescue the depressive-like behaviors of this model, as measured by the decreased immobile time in the TST and FST (Fig. 6h, i). In addition, to verify that 7, 8-DHF plays an antidepressant and anxiolytic role primarily through the lung vagal-to-brain axis, we first delivered 7, 8-DHF through intratracheal infusion, which avoided the direct uptake through the olfactory epithelium/neurons into the brain. Interestingly, the c-Fos expression peaked after 3 h of intratracheal infusion of 7, 8-DHF and then gradually decreased (Supplementary Fig. 11a, b). Moreover, our data showed that the intratracheal infusion of 7, 8-DHF could improve depressive and anxiety-like behaviors in the CRS mice. Additionally, we performed the vagotomy before intratracheal drug delivery and found that the antidepressant and anxiolytic effects of endotracheal infusion of 7, 8-DHF were significantly decreased (Supplementary Fig. 11c–g), which indicated that the vagal nerve innervating the lung might play a dominant role in 7, 8-DHF therapy. In addition, we found that direct injection of 7, 8-DHF into the DRN did not show significant antidepressant or anxiolytic effects (Supplementary Fig. 11h–l). Collectively, these data demonstrate that inhalation of 7, 8-DHF, which pharmacologically enhances TrkB activation in the lung, is a potential pathway for developing treatments aimed at convenient and effective depression therapy.

## Discussion

Our findings reveal that intravenously applied MSCs regulate the distant CNS serotonergic system via a lung vagal-to-brain axis to exert antidepressant and anxiolytic effects in mice. Furthermore, we propose a convenient inhalation treatment that aims to modulate the lung vagal-to-brain axis for depression and anxiety therapy.

The serotonin (5-HT) system has been implicated in the pathophysiology and treatment of depression and anxiety[33,34]. Selective

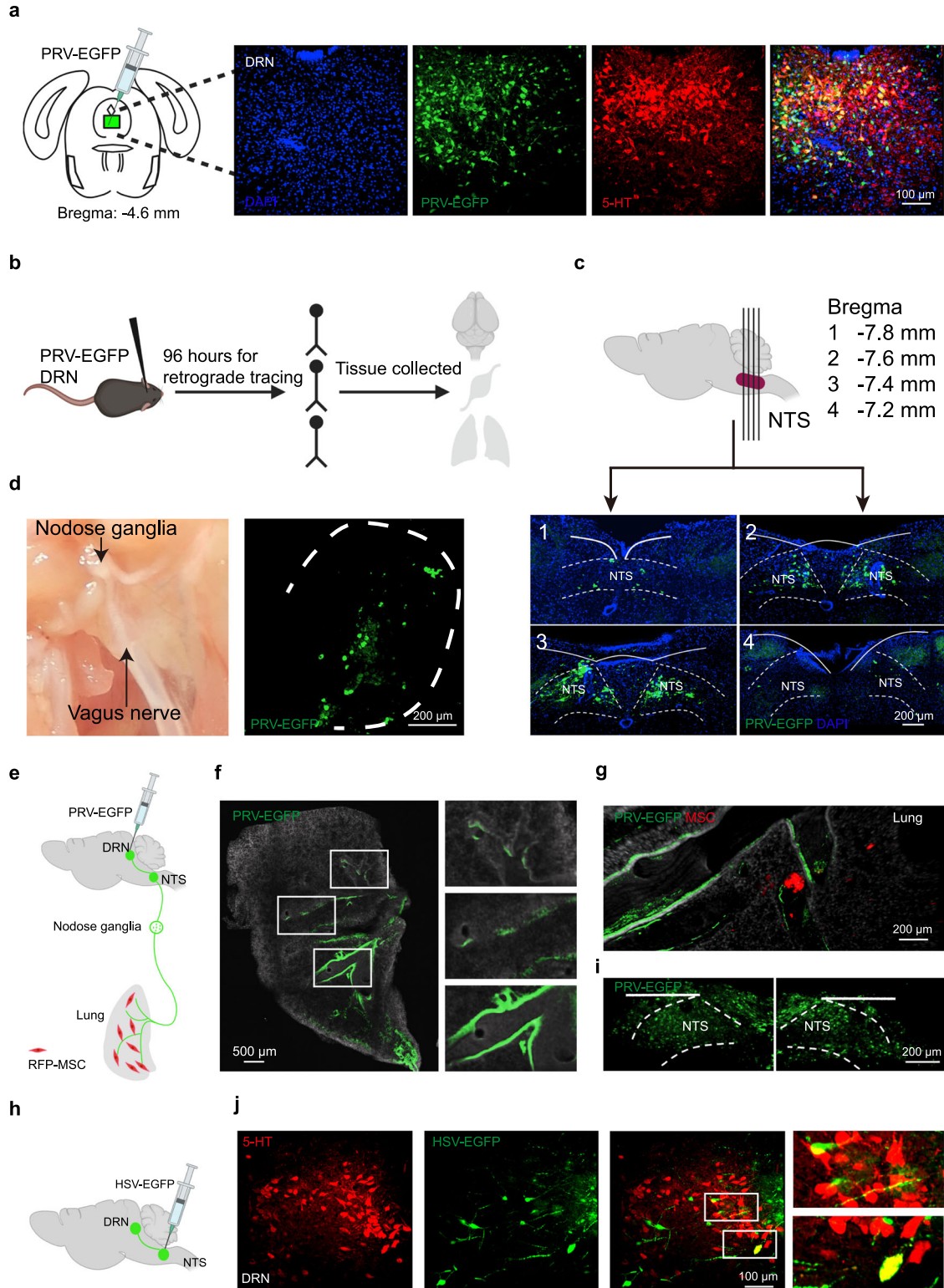

**Fig. 4 | MSCs relay neural signals to the DRN via pulmonary sensory nerves.**
**a** Polysynaptic pseudorabies PRV-GFP was injected into the DRN (Bregma: −4.6 mm). Representative images show that the infected neurons were mainly 5-HT neurons. Scale bar, 100 μm. **b** Schematic diagram showing the experimental procedures. **c−f** Dense retrograde labeling was observed in the NTS from slices 1–4 (Bregma: −7.8 mm,−7.6 mm, −7.4 mm, −7.2 mm, respectively; scale bar, 200 μm) (**c**), in the nodose ganglion (scale bar, 200 μm) (**d**), and in the lung (scale bar, 500 μm) (**f**). Schematic diagram of the retrograde transduction of PRV across synapses from the DRN region to the lung (**e**). **g** Representative images showing that RFP-MSCs were located in close proximity to PRV-GFP-marked fibers in the lungs. Scale bar, 200 μm. **h**, **i** The polysynaptic herpes simplex virus, HSV-GFP, was injected bilaterally into the NTS. Scale bar, 200 μm. **j** Representative images showing the HSV-GFP-infected 5-HT neurons within the DRN. Scale bar, 100 μm. Figure 4a, d, f–g, j are the representative images from three independent experiments. Illustrations created with BioRender.com. Source data are provided as a Source Data file.

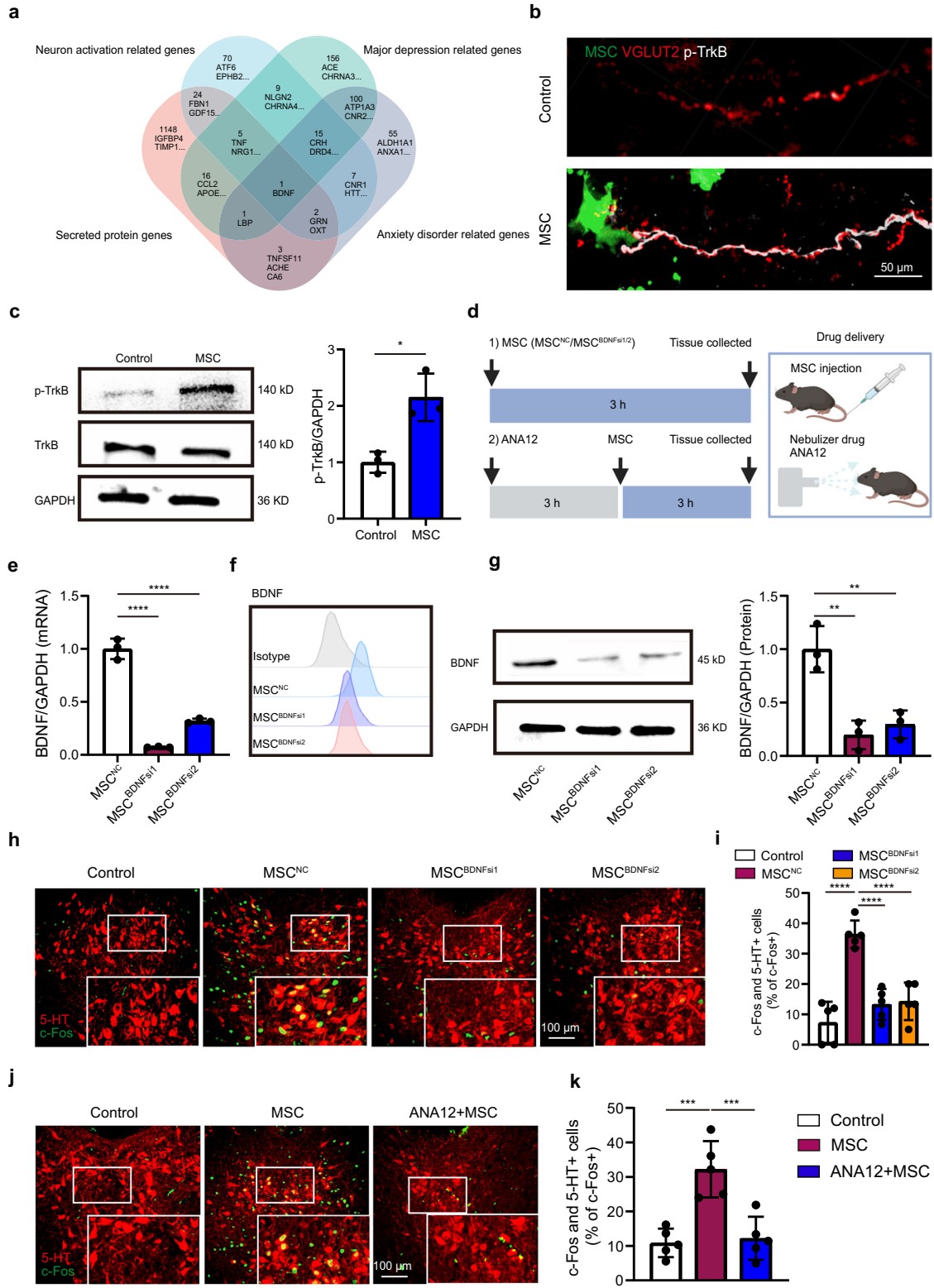

serotonin reuptake inhibitors (SSRIs) such as fluoxetine with anti-depressant and anxiolytic effects have no selectivity in augmenting 5-HT signaling, leading to adverse consequences, such as nausea, diarrhea, dyspepsia, gastrointestinal bleeding, and abdominal pain[35,36]. Our data showed that intravenously applied MSCs improved the brain serotonin level and limited depressive and anxiety-like behaviors in the mouse model but did not increase the serotonin level in the serum,

lungs, or ascitic fluid (Supplementary Fig. 12). Delayed onset is another serious limitation of classic antidepressants. A low dose of ketamine was reported to produce antidepressant effects within hours for patients with refractory depression and effectively reduce the suicidal tendencies of patients[37]. However, the potentially addictive properties of ketamine and its risk for inducing schizophrenia have raised concerns[38]. Additionally, we also observed a relatively fast-onset

**Fig. 5 | MSCs activate 5-HT neurons via brain-derived neurotrophic factor (BDNF). a** Venn diagram showing that the factors secreted by MSCs are involved in regulating neuron activation, major depression, and anxiety disorders. **b** Representative immunofluorescence images from three independent experiments showing that MSCs in the lungs activated phosphorylated TrkB (p-TrkB) in VGLUT2 nerves. Scale bar, 50 μm. **c** Representative image and quantification of western blot showing that MSC injection increased the level of p-TrkB in the lungs. Two-tailed $t$ test: $t = 4.324$, df = 4, $^* P = 0.0124$. $n = 3$ biologically independent samples. **d** Schematic diagram showing the experimental procedures used for MSC treatment and drug inhalation. qRT-PCR analysis (**e**), flow cytometric analysis (**f**),

and western blot analysis (**g**) confirming the siRNA-mediated downregulation of BDNF in MSCs. $n = 3$. **e** one-way ANOVA: $F_{(2, 6)} = 209.6$ $^{****} P = 0.0000028$. **g** one-way ANOVA: $F_{(2, 6)} = 21.07$ $^{**} P = 0.0019$. $n = 3$ biologically independent samples. **h** Representative stained images of 5-HT and c-Fos colocalization in each group. Scale bar, 100 μm. **i** Statistical analysis of 5-HT and c-Fos colocalization. One-way ANOVA: $F_{(3, 16)} = 209.6$ $^{****} P = 0.0000030$. $n = 5$ mice. **j** Representative immunofluorescence images of 5-HT and c-Fos colocalization in each group. Scale bar, 100 μm. **k** Statistical analysis of 5-HT and c-Fos colocalization. One-way ANOVA: $F_{(2, 12)} = 209.6$ $^{***} P = 0.00028$. $n = 5$ mice. Illustrations created with BioRender.com. Source data are provided as a Source Data file.

antidepressant and anxiolytic effect in CRS mice after intravenous injection of MSCs (Supplementary Fig. 13), indicating that peripheral MSCs might regulate central nervous system function. Benzodiazepines, which enhance the gamma-aminobutyric acid GABAergic system, are typical anxiolytic agents in the clinic, but numerous problems, such as cognitive or memory impairment and the rapid development of tolerance, limit their long-term use[39]. Although our data showed that MSCs may improve depressive and anxiety-like behaviors in mice, there are still some challenges to limit the clinical translation of MSC therapy, including therapeutic uncertainty, invasive procedures, and long-term safety risks[40]. Therefore, we should be cautious about MSC injection in humans, and elucidating the lung-brain axis mechanism by which MSCs exert antidepressant and anxiolytic effects in mouse models, would shed light on small compound development for mental disorders.

The DRN is a main source of 5-HT in the brain. It projects to the cortex and limbic system and plays a major role in modulating mood[41]. Moreover, the DRN has been identified as a potential therapeutic target for mental disorders in rodent models. Specifically, optogenetic stimulation of DRN Pet-1+ neurons, which are associated with 5-HT neurons reinforces mouse reward behaviors, including area-specific exploratory behavior and sucrose preference[42]. Optogenetic activation of Pet-1+ neurons in the DRN also rescues the autistic-like social deficits in Shank3-knockout mice[43]. Recently, an elegant article reported that a fast-onset antidepressant was achieved by disrupting the serotonin transporter (SERT)-neuronal nitric oxide synthase (nNOS) interaction to enhance the activity of DRN serotonergic neurons[44]. This emphasizes the importance of considering the DRN when investigating drugs for mental disorders. Overall, our study revealed a lung-originated ascending pathway and its targets in DRN. We show that intravenously infused MSCs rescued rodent depressive and anxiety-like behaviors via a lung-brain axis that mediates endogenous activation of DRN 5-HT neurons. This provides insight into a non-surgical alternative approach for targeting the DRN to treat mental disorders.

We further identified that BDNF derived from MSCs is a key messenger involved in mediating the communication between MSCs and the CNS serotonergic system via the lung vagal-to-brain axis. BDNF signaling in the CNS has long been implicated in the pathophysiology of mood disorders in humans. For example, postmortem brain samples taken from depressed patients showed decreases in the expression of BDNF and the phosphorylation/activation of the BDNF receptor, TrkB[45]. Notably, BDNF signaling is required for the actions of almost all antidepressant drugs, including SSRIs and ketamine, which are known to directly bind TrkB and allosterically increase BDNF signaling, thereby directly linking the effects of antidepressant drugs to neuronal plasticity[46,47]. In the present study, in contrast, BDNF signaling appears to act mainly as a messenger, connecting the lung to the brain via pulmonary sensory nerves. This is a role for BDNF in peripheral sensory nerves for antidepressant and anxiolytic effects. Furthermore, we propose a convenient TrkB agonist inhalation treatment that aims to stimulate the pulmonary vagal nerve for depression and anxiety therapy. Vagus nerve stimulation (VNS) is an FDA-approved therapy for the treatment of depression, but it is limited

in clinical application due to perioperative risks involved with device implantation[48]. Although transcutaneous vagus nerve stimulation (tVNS) is a non-invasive technique, the mechanism of action and influence of stimulation parameters on clinical outcomes remain predominantly hypothetical[49]. In contrast, simple and convenient TrkB agonist inhalation treatment might be an alternative strategy to VNS.

Vagus nerve innervation of peripheral organs has attracted attention as a regulator of the CNS and a potential therapeutic target in a variety of disorders, including depression and anxiety[50]. Gut-innervating vagal sensory neurons have been identified as major components of the reward circuitry, providing insight for vagal stimulation-based approaches aimed at treating affective disorders[51]. Insulin-secreting beta cells of the pancreas communicate with sensory neurons of the vagus nerve, which relay the information to nuclei of the solitary tract (NTS)[52], potentially connecting with the brain's reward system. The lung is the most extensive body surface; it is innervated mainly by vagus nerves and acts as a kind of remote warning system for the sensitive CNS[53]. Despite the accumulated knowledge, however, we know relatively little about how lung-innervating vagal sensory neuron-to-CNS signaling regulates mental disorders. In our work, we discovered a functional connection between the lung and brain, and propose this connection as a meaningful target for depression and anxiety therapy. Most importantly, we propose a convenient inhalation treatment that targets the lung-brain axis for depression and anxiety therapy.

Although the understanding of depression has advanced greatly in recent years, no single established mechanism can explain the complicated disease[54]. Here, we show that the CNS serotonergic system seems to play an important role in the antidepressant and anxiolytic effects of MSCs in mouse models and that this is mediated by the vagal lung-brain axis. Additionally, there might be other connections between lung and brain that contribute to the antidepressant and anxiolytic effects of MSCs in mice models. For example, the pulmonary microbiome could respond to microenvironmental change and thereby deliver different information to the CNS[55]. MSC-derived prostaglandin E2 could affect parasympathetic nerves innervating the lung to release the neurotransmitter, acetylcholine[56], potentially affecting the crosstalk between the lung and brain. Given that there may be various connections between lung and brain, analysis of the many bioactive molecules derived from MSCs might help us find appropriate interventions to alter the lung-brain axis for treating mental disorders. Therefore, future work should more precisely explore how the lung-brain axis responds to MSC administration under a variety of pathophysiological conditions.

In conclusion, our study reveals a functional connection between the lung and brain via pulmonary vagus nerves. More importantly, we find a pathway that can affect the lung-brain axis to relieve depressive and anxiety-like symptoms in rodents, and this further inspires an inhalation treatment that targets the vagal lung-to-brain axis for depression therapy. We believe that the precise manipulation of the lung-brain axis might shed new light on additional work in the treatment of mental disorders.

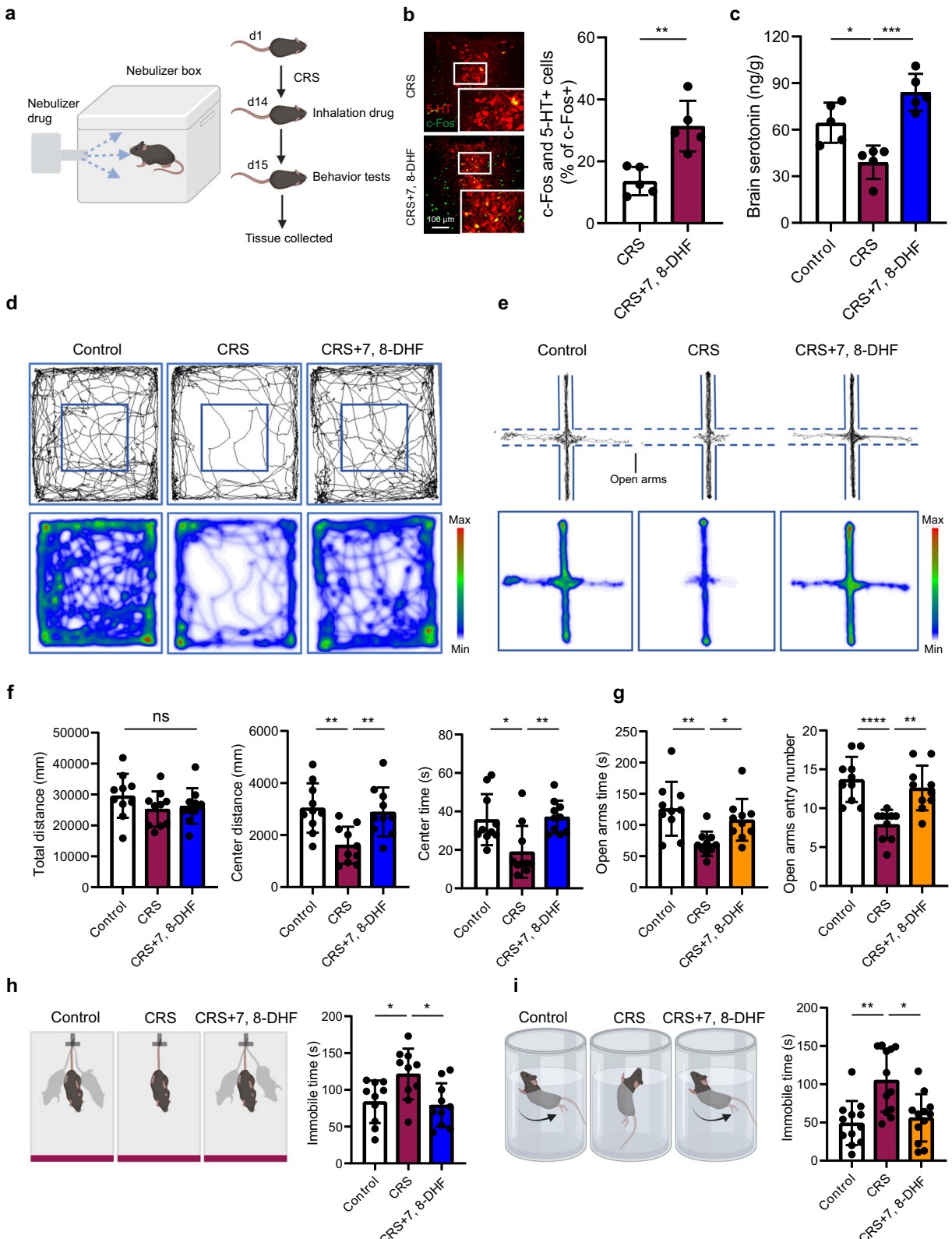

## Methods

### Animals

All animal studies and experimental procedures were approved by the Sun Yat-sen University Institutional Animal Care and Use Committee (2021001752).

Male C57BL/6 mice (6–14 weeks of age) and male CD1 mice (4–6 months of age) were purchased from the Animal Center of Sun Yat-sen University (Guangzhou, China). *VGLUT2-ires-Cre* mice (Jackson Laboratories stock No: 016963) were crossed with *GCaMP6-loxp* mice (Jackson Laboratories stock No: 029626) to obtain *VGLUT2-GCaMP6* mice (8–16 weeks of age, male), which were used for ex vivo Ca²⁺ imaging of lung peripheral sensory terminals. All mice were housed four per cage under a 12-h light-dark cycle (light on from 7 a.m. to 7 p.m., humidity between

**Fig. 6 | Inhalation of a TrkB agonist as a promising depression and anxiety treatment in CRS mice. a** Schematic diagram showing the experimental procedures related to 7, 8-DHF inhalation by CRS mice. **b** Representative immunofluorescence images showing 5-HT and c-Fos colocalization changes in the DRN after the inhalation treatment. Scale bar, 100 μm. Bar chart showing the statistical analysis of 5-HT and c-Fos colocalization in the DRN. Two-tailed $t$ test: $t = 4.256$, df = 8 ** $P = 0.0028$. $n = 5$ mice. **c** Changes in 5-HT levels in the brain homogenate supernatants after 7, 8-DHF inhalation treatment. One-way ANOVA: $F_{(2, 12)} = 18.07$ *** $P = 0.00024$. $n = 5$ mice. **d** Representative locomotion tracks and heatmaps in OFT evaluating the antidepressant effects of the 7, 8-DHF inhalation treatment in CRS mice. **e** Representative activity tracks and heatmaps in the EPM for mice of each group. **f** Statistical analysis of the total distance, the distance traveled in the center field, and the time spent in the center field of the OFT for mice of each group. Total distance, one-way ANOVA: $F_{(2, 27)} = 1.307$ ns $P = 0.29$. Center distance, one-way ANOVA: $F_{(2, 27)} = 8.054$ ** $P = 0.0018$. Center time, Kruskal-Wallis test: ** $P = 0.0066$. $n = 10$ mice. **g** Statistical analysis of the time spent in and entry numbers to open arms in the EPM for mice of each group. Open arms time, one-way ANOVA: $F_{(2, 27)} = 7.380$ ** $P = 0.0028$. Open arms entry numbers, one-way ANOVA: $F_{(2, 27)} = 13.97$ **** $P = 0.000068$. $n = 10$ mice. **h** Representative tracking of activity in the TST and statistical analysis of the duration of immobility in the TST for mice of each group. One-way ANOVA: $F_{(2, 27)} = 5.581$ ** $P = 0.0094$. $n = 10$ mice. **i** Representative schematic diagram of the FST and statistical analysis of the duration of immobility in the FST for mice of each group. Kruskal-Wallis test: ** $P = 0.0040$. $n = 12$ mice. Illustrations created with BioRender.com. Source data are provided as a Source Data file.

30% and 70%, temperatures of 20–22 °C) with free access to food and water.

### Cell sources, culture, and administration

MSCs, and human dermal fibroblasts (HDFs) used in this study have been approved by the Institutional Review Board of Sun Yat-Sen University (ZSYYL2017041).

Human cells were isolated and/or cultured as follows: for MSCs: mononuclear cells were obtained by Ficoll-Hypaque (1.077 g/mL, Amersham Biosciences) density gradient centrifugation and seeded into 75-cm$^2$ flasks (Corning) in medium consisting of low-glucose DMEM (L-DMEM, HyClone) and 10% fetal bovine serum (FBS, HyClone). After 3 days of culture, the medium was replaced and non-adherent cells were discarded. At 70–80% confluence, these cells were harvested by trypsin and cultured at $1 \times 10^4$ cells/cm$^2$ in 75-cm$^2$ flasks[57]. For HDF, a 1 cm$^2$ piece of skin was placed on culture dishes and incubated in a MF-start medium (Toyobo) for 5 days. Cells that migrated out of the graft pieces were transferred to new plates and maintained in DMEM containing 10% FBS[58]. Cells ($1 \times 10^6$ per mouse) were resuspended in 200 μL of 0.9% saline and delivered via tail-vein injection.

### Chronic restraint stress (CRS) and MSC treatment

Mice were subjected to restraint stress by being placed in well-ventilated 50-ml polypropylene conical tubes for 4–5 h per day for 14 consecutive days. After each day of restraint stress, mice were returned to their home environment, where they were housed in normal plastic cages with free access to food and water[59]. On day 11 of restraint stress, mice in the treatment group received MSCs via tail-vein injection, mice in the restraint group received 0.9% saline as an injection control and unstressed control animals remained undisturbed.

### Repeated social defeat (RSD) and MSC treatment

Male C57BL/6 mice (6–8 weeks old) were individually exposed for 2 h to a CD1 male mouse (4–6 months old) each day for up to 6 consecutive days. During RSD sessions, CD1 males displayed dominant behaviors and mounted frequent attacks on C57B/L6 mice. A CD1 male mouse that failed to mount an attack within the first 5 min was replaced with a more aggressive CD1 male mouse. The CD1 male mice were rotated such that each C57BL/6 mouse was never exposed to the same aggressor on consecutive days. C57BL/6 mice displayed submissive behaviors, including fleeing, rearing, and huddling, throughout RSD and were closely monitored for signs of injury. Injured animals (<2%) were removed from RSD and did not contribute to the data generated. On day 2 of RSD, mice in the treatment group received MSCs via tail-vein injection, mice in the RSD group received 0.9% saline as an injection control, and unstressed control C57BL/6 mice remained undisturbed in their home cages with no exposure to CD1 male mice.

### Behavioral assessments

All behavioral tests were performed during the light cycle in a dedicated sound-proof behavioral facility; assessments were performed by experimenters blind to treatment. Mice were brought to the testing room 30 min before the start of each behavioral test and remained in the same room throughout the test. After each behavioral test, the instruments were cleaned with 75% ethanol.

### Open field test (OFT)

Mice were placed in the open field chamber (50 cm × 50 cm × 50 cm), which was divided into a central field (25 cm × 25 cm) and a peripheral field. Mouse behaviors were monitored for 10 min using a SONY HDR-CX405 video camera. The total distance, the distance traveled in the center field, and the time spent in the center field were analysed by behavioral analysis software (TopScanLite, Clever System, USA).

### Elevated plus maze (EPM)

The elevated plus maze consisted of two opposite open arms (28.5 cm × 7 cm), two opposite closed arms (28.5 cm × 7 cm), and one central zone (7 cm × 7 cm). The plus-shaped apparatus was elevated 55 cm above the ground. During the test, mice were placed in the open arm of the maze, facing the center zone, and a 10-minute monitoring of behavior was performed using a SONY HDR-CX405 video camera. The time and entry numbers in the open arms were quantified and analysed by behavioral analysis software purchased from Clever Systems.

### Tail suspension test (TST)

Mice were suspended by the tail, which was taped and secured to a horizontal bar 30 cm above the floor, which ensured that the mice could not make any other contact or climb during the assay. Animal behaviors were videotaped for 6 min using a SONY HDR-CX405 video camera. The animals were habituated for the first 1 min, and the time spent immobile during the subsequent 5-minute test period was counted offline by an observer blinded to animal treatment.

### Social avoidance test

A social avoidance test was performed under red-light conditions. Firstly, C57BL/6 mice were placed in an open field (40 cm × 40 cm) with a small wire animal cage placed at one end and filmed from above for 150 s using an elevated, tripod-mounted SONY HDR-CX405 video camera. Next, each C57BL/6 mouse was placed in the open field for an additional 150 s with an unfamiliar CD1 mouse (target mouse) placed inside the small wire animal cage. Social interaction was defined as the presence of the C57BL6 mouse within an interaction zone (24 cm × 14 cm) surrounding the wire animal cage. The sociability index was calculated by dividing the time spent in the interaction zone when the CD1 mouse was present by the time spent in the interaction zone when the CD1 mouse was absent. C57BL/6 mice with a sociability index below 100 were classified as susceptible and those with a sociability index above or equal to 100 were classified as resilient.

### Forced swimming test (FST)

Mice were placed in a glass cylinder (height: 30 cm, diameter: 16 cm) containing water maintained at 24 °C. The depth was set at 14 cm, and the mice could not escape or touch the bottom. Mice were forced to

swim for 5 min, and were videotaped using a SONY HDR-CX405 video camera. The animals were habituated for the first 1 min and their behaviors were analysed over the next 4 min. The time spent immobile during the 4-minute test period was counted offline by an observer blinded to animal treatment.

## Stereotaxic injection

For intracranial injection, mice were anaesthetized with isoflurane (3% during induction, 1.5% during maintenance, flow rate = 2 L/min) and fixed on a stereotaxic device (RWD Life Science) equipped with an electronic heating pad. The scalp was cut and a 10-mm incision was made posterior to the bregma along the midline. Injections were performed with a Hamilton 1.0-μL syringe needle (Hamilton 7000 series) and a microsyringe pump (KDS Legato™ 100, Sigma-Aldrich) at a rate of 100 nL/min. After injection, the glass pipette was left at the site for 5 min and then slowly withdrawn. The scalp was sutured and the mouse was allowed to fully recover from anesthesia and returned to its home cage. All viral vectors and drugs were subdivided into aliquots and stored at −80 °C until use.

## In vivo electrophysiology (DRN) and data analysis

For in vivo electrophysiology recording, a custom-made electrode (Yige Biotechnology Co., Ltd., Jiangsu, China) consisting of four tetrodes was inserted slowly towards the DRN (AP: −4.6 mm; ML: 1.1 mm; DV: −3.2 mm, at a 20° angle lateral to the midline) of mice. Three screws were embedded in the skull and stainless steel wires were wound around the skull-penetrating screws as a ground. The electrode was fixed to the skull with dental cement. After surgery, the mice were individually housed for at least 1 week. After recovery, the mice were allowed to adapt to the recording head stage for 10 min per day for 2–3 days. Local field potential (LFP; digitized at 1 kHz, low-pass filtered up to 250 Hz) and spontaneous spiking activity (digitized at 40 kHz, band-pass filtered between 300 and 6000 Hz) were recorded. Data were recorded for 30 min before (baseline) and after MSC treatment (approximately 1 h after MSC injection). 5-HT neurons exhibit a slow (0.1–4 Hz) and regular firing rate (coefficient of variation [COV], ranging from 0.12 to 0.87) and a broad biphasic (positive–negative) or triphasic waveform (0.8–3.5 ms; 1.4 ms first positive and negative deflections)[60]. All experiments were conducted during the light phase. All data recorded from each microelectrode were imported into Offline Sorter V4 (Plexon Inc.). Single units were manually identified by threshold-crossing and principal component analysis (PCA). Data analysis was conducted using Neuroexplorer 5 (Plexon Inc.) and MATLAB. The average firing rates before and after MSC injection were statistically analysed by the two-tailed $t$ test.

## Patch-clamp recording

The patch-clamp recording technique was used to measure spontaneous action potential frequencies in DRN 5-HT neurons. Visualization of 5-HT neurons was through mice injected with rAAV-TPH2-mCherry-WPRE-hGH (purchased from BrainVTA (Wuhan, China)) into the DRN (Anterior-posterior (AP): −4.6 mm; Medial-lateral (ML): −1.1 mm; Dorsal-ventral (DV): −3.2 mm; at a 20° angle lateral to the midline). The injection volume of AAV per mouse was 500 nL. The stereotaxic injection was performed as described above. Next, DRN containing coronal slices (300 μm) were cut with a Leica VT1200s vibratome, recovered at 34 °C for 10–13 min in NMDG cutting solution (93 mM NMDG, 93 mM HCl, 2.5 mM KCl, 1.2 mM NaH$_2$PO$_4$, 30 mM NaHCO$_3$, 25 mM D-glucose, 20 mM HEPES, 5 mM Na-ascorbate, 2 mM thiourea, 3 mM Na-pyruvate, 10 mM MgSO$_4$, and 0.5 mM CaCl$_2$, pH 7.35 with NMDG or HCl), and then maintained at 25 °C in oxygenated artificial cerebrospinal fluid (ACSF: 124 mM NaCl, 3 mM KCl, 2 mM CaCl$_2$ 1.3 mM MgCl$_2$, 25 mM NaHCO$_3$, 1.25 mM NaH$_2$PO$_4$, and 10 mM glucose) for 1 h until electrophysiological recordings. Slices were transferred to the

recording chamber and superfused with oxygenated ACSF (3 mL/min). Slices were visualized with infrared optics using an upright microscope equipped with a 60x water-immersion lens (Nikon E600). Neurons infected with rAAV-TPH2-mCherry in the DRN were identified by their location and fluorescence. Recording pipettes were filled with a solution containing: 135 mM K-gluconate, 5 mM KCl, 0.5 mM CaCl$_2$, 10 mM HEPES, 2 mM Mg-ATP, 0.1 mM GTP, 5 mM EGTA, and 300 mM mOsm (pH adjusted to 7.3 with KOH). Current-evoked action potentials (APs) were elicited with 500-ms current injections at eight intensities (0, 20, 40, 60, 80, 100, 120, and 140 pA), with a 30-s trial interval. All signals were acquired with a MultiClamp 700B amplifier (Molecular Devices), filtered at 2 kHz, and sampled at 10 kHz with a Digidata 1440 A interface using Clampex 10.2 (Molecular Devices). Data were accepted when series resistance fluctuated within 15% of the initial values (15–25 MΩ). Action potential frequencies were analysed using Clampfit 11.2 (Molecular Devices).

## Fiber photometry recording

*GCaMP6-loxp* mice were injected with rAAV-TPH2-CRE-WPRE-hGH and the DRN was implanted with an optical fiber (0.2 mm O.D., 0.37 mm numerical aperture (NA); Inper Ltd., China) with a ceramic ferrule (AP: −4.6 mm; ML: 1.1 mm; DV: −3.2 mm, at a 20° angle lateral to the midline). The ceramic ferrule was supported with a screw and dental cement. Mice were housed for 3 weeks to allow for virus expression and recovery, and then a fiber photometry system (Nanjing Thinkertech) was utilized to record calcium activity in the DRN. According to the manufacturer's instructions, the 473-nm excitation light was reflected off a dichroic mirror and focused by a 10-objective lens to excite GCaMP6 fluorescence. Calcium fluorescence signals were collected by a photomultiplier tube and converted into electrical signals. The GCaMP6 fluorescence signals in the DRN were recorded before and after MSC injection. Photometry data were exported as MATLAB files for further analysis. Changes in fluorescence intensity were expressed as percentage changes over baseline (($F$-$F_0$)/$F_0$). F means were determined from the average fluorescence intensity during recording.

## 5, 7-Dihydroxytryptamine (5, 7-DHT) lesion

The elimination of serotonergic neurons in the DRN (dorsal raphe nucleus) was performed via stereotaxic injection of 5, 7-DHT (a selective serotonergic neurotoxin)[61]. Briefly, 5 mg of 5, 7-DHT (Sigma Aldrich) was dissolved in 1.67 mL of 0.9% saline containing 0.1% ascorbic acid to yield a 3 μg/μL solution, which was stored at −80 °C until use. The injection volume was 2 μL per mouse. The drug injection was performed as described in the stereotaxic injection section of the Method section.

## AAVs for 5-HT neuron elimination

To specifically eliminate the serotonergic neurons in the DRN, a 1:1 mixture of rAAV-TPH2-CRE-WPRE-hGH and rAAV-EF1a-DIO-taCasp3-TEVp-WPRE-hGH purchased from BrainVTA (Wuhan, China) was injected into the DRN (Anterior-posterior (AP): −4.6 mm; Medial-lateral (ML): −1.1 mm; Dorsal-ventral (DV): −3.2 mm; at a 20° angle lateral to the midline). The injection volume of the AAV mixture per mouse was 500 nL.

## AAVs for hM4Di-mediated DRN 5-HT neuron inhibition

To specifically induce the silencing of DRN 5-HT neurons, a 1:1 mixture of rAAV-TPH2-CRE-WPRE-hGH and rAAV-EF1a-DIO-hM4D(Gi)-mCherry-WPRE-hGH (BrainVTA, Wuhan, China) was injected into the DRN. The injection volume of the AAV mixture per mouse was 500 nL. After 3 weeks, the DREADD agonist clozapine N-oxide (CNO) was injected intraperitoneally (1 mg/kg) one hour before the MSC injection. Mice injected with AAV and saline (instead of CNO) were used as the vehicle group.

## PRV-EGFP for retrograde tracing

PRV-EGFP purchased from Braincase (Shenzhen, China) was injected in the DRN (Anterior-posterior (AP): −4.6 mm; Medial-lateral (ML): −1.1 mm; Dorsal-ventral (DV): −3.2 mm; at a 20° angle lateral to the midline) for retrograde projection mapping. The injection volume per mouse was 500 nL.

## HSV-EGFP for anterograde tracing

HSV-EGFP purchased from Braincase (Shenzhen, China) was injected into the NTS (Anterior-posterior (AP): −7.5 mm; Medial-lateral (ML): +0.3/−0.3 mm; Dorsal-ventral (DV): −5 mm) for anterograde projection mapping. The injection volume per mouse was 500 nL.

## Unilateral vagotomy

A longitudinal midline incision was made in the ventral region of the neck before blunt dissection. The overlying muscles and fascia were separated to reveal the left or right vagal nerves[62]. For the vagotomy group, the vagus was carefully stripped away from the carotid artery and precisely cut. For the sham group, the vagus was kept intact. The wound was closed and sutured.

## Immunofluorescence analysis

Mice were transcardially perfused with PBS followed by 4% paraformaldehyde. Following perfusion, tissues were harvested and kept in 4% paraformaldehyde for 6−24 h and then in 30% sucrose solution for 24−48 h. The tissues were then frozen and cut into 40-μm sections. The tissue sections were incubated with 5% normal goat serum (Sigma) and 0.1% Triton X-100 (HyClone) in PBS for 1 h at room temperature, followed by incubation with primary antibodies at 4 °C overnight. Primary antibodies included (1) mouse anti-c-Fos IgG, 1:1000, mouse monoclonal (2H2) to c-Fos, Abcam, Catalog No. ab208942; (2) rabbit anti-5-HT, 1:500, Immunostar, Catalog No. 20080; (3) rabbit anti-Iba-1, 1:1000, FUJIFILM Wako Pure Chemical Corporation, Catalog No. 019-19741; (4) rabbit anti-TH, 1:1000, Millipore, Catalog No. AB152; (5) rabbit anti-CRH, 1:500, Immunostar, Catalog No. 20084; (6) rabbit anti-Tubb3, 1:500, Abcam, Catalog No. ab18207; (7) rabbit anti-VGLUT2, 1:300, rabbit monoclonal (EPR21085) to VGLUT2, Abcam, Catalog No. ab216463; (8) rabbit anti-Phospho-TrkA+TrkB, 1:250, Bioss, Catalog No. bs-3457R.

Next, the sections were incubated with secondary antibodies at room temperature for 1 h. Secondary antibodies included (1) goat anti-mouse IgG Alexa 488, 1:1000, Invitrogen, Catalog No. A-11001; (2) goat anti-rabbit IgG Alexa 488, 1:1000, Invitrogen, Catalog No. A-11008; (3) goat anti-rabbit IgG Alexa 594, 1:1000, Invitrogen, Catalog No. A-11012; (4) goat anti-rabbit IgG Alexa 647, 1:1000, Invitrogen, Catalog No. A-21245. Additionally, nuclei were counterstained with DAPI (4′, 6-diamidino-2-phenylindole) for 10 min. The primary and secondary antibodies are also listed in the Supplementary Information Table 2. Images were acquired using a Dragonfly CR-DFLY-202 2540 confocal microscope, LEICA DMi8 microscope, and Nikon Eclipse Ni-E confocal microscope.

## c-Fos measurements

To determine the effects of MSCs on brain neuronal activity, mice were sacrificed 3 h after MSC injection and perfused as described above. Brain frozen sections were obtained and the expression of c-Fos was detected by immunofluorescence to visualize neuronal activation in the dorsal raphe nucleus (DRN), ventral tegmental area (VTA), paraventricular nucleus (PVN), and nucleus tractus solitarius (NTS). To detect the activation of neurons in the nodose ganglion, mice were sacrificed 90 min after MSC injection and perfused as described above, and the vagus nerve was separated from the carotid artery until the nodose ganglion became accessible. To visualize neuronal activation, the expression of c-Fos was detected by immunofluorescence of the whole nodose ganglion. The utilized primary and secondary antibodies are listed in the Supplementary Information.

## Enzyme-linked immunosorbent assay (ELISA)

Mice were anaesthetized with isoflurane at the indicated time points and blood was collected via cardiac puncture and centrifuged at $600 \times g$ for 15 min to obtain serum. Fresh mouse brains were removed, homogenized in PBS containing protease and phosphatase inhibitors (Roche), and centrifuged at $10,000 \times g$ for 10 min. The supernatant was collected and immediately stored at −80 °C until use. Commercial ELISA kits were used for quantitative detection of IL-6 (Invitrogen, 88-7064), TNF-α (Invitrogen, 88-7324), 5-HT (Abcam, ab133053), dopamine (Novus, NBP2-67270), and CRH (Finetest, EM0465).

## High-performance liquid chromatography (HPLC) for 5-HT detection

Fresh mouse brains were removed, homogenized in PBS containing protease and phosphatase inhibitors (Roche), and then centrifuged at 10,000 g for 10 min. The supernatant was collected as the brain homogenate, and 100 μL of brain homogenate per mouse was transferred to a 1.5-mL microcentrifuge tube, combined with 300 μL methanol containing 0.1% formic acid, vortexed for 5 min, and centrifuged at $15,000 \times g$ for 10 min. The supernatant was collected for HPLC detection. All samples were run on a liquid chromatograph (UltiMate 3000 RS) and detected by tandem mass spectrometry (UltiMate 3000 RS).

## Flow cytometry

A CytoFLEX flow cytometer was used to perform flow cytometry experiments, and the results were assessed with the CytExpert (Beckman) and FlowJo X 10.0.7r2 software packages (BD). The gating strategy is provided in the Supplementary Fig. 14. The antibodies utilized for flow cytometry are listed in Supplementary Information Table 3. Specifically, (1) rabbit anti-BDNF, 1:30, rabbit monoclonal (EPR1292) to BDNF, Abcam, Catalog No. ab108319; (2) goat anti-rabbit IgG Alexa 594, 1:1000, Invitrogen, Catalog No. A-11012.

## Establishment of mCherry-GFP-MSCs, GFP-MSCs, and RFP-MSCs

According to a method previously published in an elegant article[63], mCherry-GFP-MSCs were constructed using EGFP and a modified mCherry lentiviral expression vector. Briefly, the sLP-mCherry sequence and EGFP sequence were cloned into a pRRL lentiviral backbone. A soluble peptide (SP) and a modified TAT peptide were cloned upstream of the mCherry cDNA (sLP-mCherry). MSCs were stably infected with lentiviral particles. mCherry-GFP-MSCs, which were identified as those overexpressing EGFP and secreting extracellular vesicles expressing mCherry, could be applied to visualize the paracrine effects of MSCs on neighboring cells in vivo. GFP-MSCs and RFP-MSCs were transduced by separately introducing EGFP and RFP lentiviral expression vectors. The relevant sequence information is presented in the Supplementary Information.

## Confocal imaging of ex vivo living lungs (*VGLUT2-GCaMP6* mice)

Fresh lungs were dissected from *VGLUT2-GCaMP6* mice, and placed in a perfusion imaging chamber (Warner Instruments) filled with physiological buffer (125 mM NaCl, 5.9 mM KCl, 2.56 mM CaCl$_2$, 1 mM MgCl$_2$, 25 mM HEPES, 0.1% BSA, pH 7.4), and imaged on a Vagus nerve innervation pus FVMPE-RS confocal microscope. MSCs were prepared in suspension ($1 \times 10^6 / 200$ μL of 0.9% saline) and added to the perfusion imaging chamber. The GCaMP6 fluorescence intensity of each frame was quantified by Image J software (https://imagej.nih.gov/ij/). Changes in fluorescence intensity are expressed as the percentages of change over baseline $((F - F_0)/F_0)$.

## Drug inhalation treatment

To target the lungs, the TrkB antagonist, ANA12 (0.5 mg/kg), and the TrkB agonist, 7, 8-DHF (10 mg/kg), were delivered via inhalation treatment. 7, 8-DHF was purchased from Tokyo Chemical Industry and prepared in a vehicle of 17% dimethylsulfoxide in PBS. ANA-12 was purchased from MedChemExpress and prepared in a vehicle of 1% dimethylsulfoxide in physiological saline.

## RNA isolation and quantitative real-time and reverse transcription PCR

Total RNA was extracted using the TRIzol reagent (Invitrogen), and 1 μg of RNA was reverse transcribed using a RevertAid First Strand cDNA Synthesis Kit (Thermo Scientific). The generated cDNA was subjected to real-time PCR with the SYBR Green reagent (Roche). The primer sequences are listed in the Supplementary Information.

## RNA interference (RNAi)

To knock down brain-derived neurotrophic factor (BDNF) in MSCs, BDNF interference vectors were synthesized from RIBOBIO (Guangzhou, China). MSCs were seeded into six-well plates and transfected with interference vectors using Lipofectamine™ RNAiMAX reagent (Invitrogen). The RNAi sequences are presented in the Supplementary Information.

## Western blot assays

Fresh mouse lungs and DRNs were collected for western blot analysis. The DRN was obtained from 1-mm-thick fresh mouse brain coronal sections kept on wet ice. Tissues were homogenized in RIPA lysis buffer (Sigma) containing phosphatase and protease inhibitor cocktails (Roche) and then centrifuged at 10,000 rpm for 15 min at 4 °C. MSC lysates were prepared with RIPA lysis buffer (Sigma) containing phosphatase and protease inhibitor cocktails (Roche) and centrifuged at 4 °C. In both cases, supernatants were collected and immediately stored at −80 °C until use. Proteins were separated by SDS-PAGE, transferred to a polyvinylidene fluoride (PVDF) membrane, blocked with 5% non-fat milk, incubated overnight with the appropriate primary antibody at 4 °C, and then incubated with secondary antibodies at room temperature. All of the utilized antibodies are listed in the Supplementary Information Table 4. Specifically, primary antibodies included (1) mouse anti-c-Fos IgG, 1:1000, mouse monoclonal (2H2) to c-Fos, Abcam, Catalog No. ab208942; (2) rabbit anti-BDNF, 1:1000, rabbit monoclonal (EPR1292) to BDNF, Abcam, Catalog No. ab108319; (3) rabbit anti-TrkB, 1:1000, rabbit monoclonal (80E3) to TrkB, Cell Signaling Technology, Catalog No. 4603 S; (4) rabbit anti-Phospho-TrkA+TrkB, 1:1000, Bioss, Catalog No. bs-3457R; (5) rabbit anti-GAPDH, 1:2000, rabbit monoclonal (14C10) to GAPDH, Cell Signaling Technology, Catalog No. 2118 S. Secondary antibodies included (1) anti-rabbit IgG, HRP-linked antibody, 1:2000, Cell Signaling Technology, Catalog No. 7074 S; (2) anti-mouse IgG, HRP-linked antibody, Cell Signaling Technology, Catalog No. 7076 S. Antigen-antibody complexes were detected by enhanced chemiluminescence (GE Healthcare).

## RNA sequencing

MSC samples were used for the bulk RNA-seq analysis. TRI Reagent (TR118, Molecular Research Center, Inc.) was used to extract total RNA according to the manufacturer's protocol. RNA samples with an RNA integrity number >8 were used to prepare libraries, and paired-end sequencing was performed by GENEWIZ (China) following a standard protocol. Raw sequencing data were demultiplexed and filtered using bcl2fastq2 and fastp. The data were then mapped to the Human Genome hg38 reference sequences, and the gene expression values were estimated using CLC Genomics Workbench 11.0 (Qiagen). The similarity analysis of the whole secreted proteins of MSCs was performed based on the Pearson correlation coefficient (R).

The raw bulk RNA-seq data in our study was processed using the established protocol provided by Illumina. Subsequent data analysis was conducted using CLC Genomics Workbench 11.0, a commercially available software solution.

## Statistics and Reproducibility

All summarized data are expressed as the mean ± SD (see Figs. 1d, f–g, i–j, l, n, q–r; 2b–c, e, g, i–j, l–m; 3b, h, l, n–o; 5c, e, g, i, k; 6b–c, f–i). In the box plot (Fig. 1b), lower and higher bounds, central line, lower and higher whiskers respectively represent 25th, 75th, 50th, 5th, and 95th percentiles. Statistical comparisons were made with Prism software (v 8.02, GraphPad) using the two-tailed Student's $t$ test (between two groups) or the one-way ANOVA (for multigroup comparisons) as appropriate. Before performing one-way ANOVA or $t$ test, the normality and homogeneity of variances were tested. When the normality assumption is violated, we performed the Kruskal-Wallis test (the nonparametric equivalent of the one-way ANOVA) or the Mann-Whitney U test. Moreover, when the homogeneity of variances is violated, we performed the Brown-Forsythe ANOVA test.

Detailed information on statistical tests and sample sizes is indicated in the figure legends. All experiments involved biological (not technical) replicates. Specifically, in behavioral tests, fiber photometry recording, immunofluorescence analysis, and ELISA detection, n indicates mouse number; in qRT-PCR and WB assays, n indicates biologically independent samples; in electrophysiology recording, n indicates neuron number from three independent experiments. All representative micrographs are from three independent experiments. *$P < 0.05$, **$P < 0.01$, ***$P < 0.001$, ****$P < 0.0001$, ns - no significant difference. Source data in all figures are provided in a Source Data file submitted with this paper. The sample sizes, specific statistical tests used, and main effects of our statistical analyses for each experiment are reported in the Source Data file.

## Reporting summary

Further information on research design is available in the Nature Portfolio Reporting Summary linked to this article.

## Data availability

The authors declare that all data supporting the results of this study are available within the paper and its Supplementary Information. The datasets generated and/or analysed during the current study are available from the corresponding author upon request. The raw sequence data reported in this paper has been deposited in the Genome Sequence Archive with ID HRA005064 and Gene Expression Omnibus GSE246290. Source data are provided with this paper. The dataset generated and analysed during the current study is available in the Figshare repository https://doi.org/10.6084/m9.figshare.23660643. Source data are provided with this paper.

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

## Acknowledgements

This study was supported by the National Key Research and Development Program of China, Stem Cell and Translational Research (2022YFA1104100, 2019YFA0110300, 2019YFA0110303), National Natural Science Foundation of China (32130046, 82170540, 81970109, 81721003), Natural Science Foundation of Guangdong Province (2018A0303130305, 2021A1515011759, 2022A1515012452), Pioneering talents project of Guangzhou Development Zone (2021-L029), Key Research and Development Program of Guangdong Province (2019B020234001, 2019B020236002), Science and Technology Program of Guangzhou (202206080002, 2023B01J1002, 2023A04J2063).

## Author contributions

J.H. conceived, performed, and analysed the experiments in this study. J.Z.Y. and Y.W.D. conceived and performed behavioral tests and stereotaxic injection together. W.J.H. and R.J.L., generated and analysed RNA sequencing data. J.Y.C. and Y.Q. performed brain section staining and microscopy under the supervision of X.Y.C. J.H.S. T.W. contributed confocal images of MSC distribution and discussed results. A.P.X. and X.R.Z. conceived and designed the study. A.P.X., X.R.Z., and J.H. wrote the manuscript with input from J.Z.Y. and W.J.H. A.P.X. and X.R.Z. supervised the whole project.

## Competing interests

The authors declare no competing interests.

## Additional information

**Peer review information** : *Nature Communications* thanks Tomas Huerta, Chian-Yu Peng and the other, anonymous, reviewers for their contribution to the peer review of this work. A peer review file is available.

