## [Peer Review File · Nature Communications]

Mesenchymal Stromal Cells Alleviate Depressive and Anxiety-like Behaviors via a Lung Vagal-to-Brain Axis in Male MiceREVIEWER COMMENTS

Reviewer #1 (Remarks to the Author):

This manuscript by Huang et al. describes a previously uncharacterized mechanism by which intravenous infusion of mesenchymal stem cells attenuate stress and anxiety-associated behavior through pulmonary vagal nerve mediated serotonin circuit stimulation. Although vagus nerve stimulation is already an FDA-approved treatment for major depressive disorders (MDD), additional information regarding its' mechanism remains elusive. This study provides an intriguing potential point of interaction between pulmonary and neural interactions in the context of MSC infusion treatment of MDD. I find this manuscript clearly organized and well written, with appropriately controlled experiments that tested the key components of the hypothesis. However, I would like to see the follow points addressed before recommending this manuscript for publication at Nature Communications.

Major points:

1. Despite the visualization of GFP+ cells in the lung, it is unclear whether the observed GFP+ cells retain MSC properties at the time of analysis. The authors should include immunohistochemical confirmation of MSC specific markers and the lack of markers of cell lineages (ie bone or adipocytes) that MSCs are capable of generating. This analysis is necessary to determine the source of secreted factors that contribute to the antidepressant effect.
2. The authors previously published that MSC-induced cholinergic anti-inflammatory signaling could attenuate the effect of LPS induced lung inflammation. Although the authors did not observe changes in IL6 and TNF α levels after stressor, the increase in IL6 after CRS or RSD is well-established in the literature. The absence of change in IL6 here is likely due the difference in measuring at the acute (4 hrs) vs. chronic (11 days) time points after stressors. To properly assess the influence of MSC infusion on IL-6 and TNF-alpha levels, the experimental design needed to be adjusted to capture the increase in IL-6 post stress and determine whether the presence of MSC alters this response. Alternatively, downstream signaling (ie pSTAT3) or transcriptional targets (ie SOCS3) or physiological readout (ie reactive gliosis) of these signaling pathways should also be examined at the experimental end point to conclude that MSC infusion does not influence depressive behavior through effects on stressors-induced inflammatory responses.
3. In addition to c-Fos staining, the electrophysiological recordings in panels 1o-1q provide additional support for MSC induced neuronal activation in DRN. However, controls with no MSC infusion and with fibroblast infusion should be included for the experiment to allow the interpretable results.
4. I appreciate the authors used the two independent methods to ablate of 5-HT expressing neurons and find the results supporting that notion that 5-HT neurons are necessary for the MSC induced anti-depressant effect. However, the point about MSC inducing activation of 5-HT neurons can be more convincingly addressed with hM4Di silencing of 5-HT+ neurons in the presence of MSC infusion. Similarly, observation of 5-HT cell activation with virally delivered, hM3Dq-mediated vagus nerve stimulation would also strengthen this important point.
5. 7,8-DHF crosses the blood-brain barrier and intranasal delivery of the agonist likely allow direct uptake of the drug through the olfactory epithelium/neurons into the brain through retrograde transport, thus

the experimental approach does not limit effect of the drug specifically to vagal nerve endings in the lung. To support the hypothesis, the authors need to provide data that demonstrate the agonist's effects are through TrkB activation in the lung and not directly in the brain.

Minor points:

1. Although briefly described in the Methods section, specific statistical analyses used for each experiment should be included in the figure legends.
2. Increased hippocampal neurogenesis is frequently observed with anti-depressant action and is known to be promoted by elevated BDNF levels. It would be interesting for the authors to determine whether improved affective behavior is associated with changes in hippocampal neurogenesis.
3. From a clinical application perspective, the safety and efficacy of intravenous mesenchymal infusion remain not fully understood. With the availability of VNS for MDD, it is difficult to imagine a scenario which MSC infusion pose significant advantage over VNS. While intranasal delivery of TrkB agonist is proposed as an alternative treatment strategy, discussion on the advantages and limitations of the approach in comparison to VNS should be included.

Reviewer #2 (Remarks to the Author):

In this study, Huang and colleagues investigated the antidepressant effect of mesenchymal stromal cells (MSCs) in chronic restraint stress (CRS) mouse model of depression and anxiety. The authors showed that intravenous injection of MSCs displays significant antidepressant effects, evidenced by reversal efficacy of CRS-induced deficits in open field test (OFT), elevated plus maze (EPM) and tail suspension test (TST). They further showed that iv MSCs exert their effect through lung-vagal sensory neurons-NTS-DRN 5-HT neurons. At molecular level, they provide evidence supporting that BDNF-TrkB signaling plays important role in mediating MSCs-induced antidepressant effect.

Overall, this is a very interesting study. Experiments are well executed and analyzed. However, there are some concerns that need to be thoroughly addressed.

1. Abbreviation HDF in Figure 1a should be explained in figure legend (human dermal fibroblast)
2. In Figure 1b, it is confusing what the top horizontal line indicates. One line should indicate the comparison between Control and CRS, and another one for comparison between CRS and MSC? Additionally, the labeling for groups is also confusing. My understanding is that Control, CRS, CRS+HDF, and CRS+MSC. If this is the case, the authors should provide injection groups of HDF alone and MSC alone in control mice (stress-naïve mice).

3. The in vivo recording data are interesting. However, it is not clear what kind of mice they used to record DRN neurons: stress-naïve control or CRS-treated mice? It would be important to perform recording from both control and CRS neurons. Additionally, it is also unclear how the DRN 5-HT neurons were identified in those in vivo recordings, and what time after MSC treatment the recordings were performed. An interesting question is how long time the effect on DRN neurons lasts.

4. In general, n for some experiments is low (Figure 1n, 1q, 1r). For Figure 1r, what is MSC group: MSC infusion alone or CRS + MSC infusion? It would be important for the authors to provide CRS + MSC infusion group. One interesting question is what the possible mechanism for the significant decrease of 5-HT in the CRS group of Figure 1r is. It would be useful to explore the intrinsic excitability of DRN 5-HT neurons in control, CRS, and CRS + MSC infusion groups.

5. The results in Figure 1q showed that the fiber photometry recordings were obtained from DRN 5-HT neurons of GCaMP6-loxp mice. Validation data would be needed to show GCaMP6 expression selectively in 5-HT neurons.

6. It is unclear what is 7,8-DHF group in Figure 6. Is it 7,8-DHF alone or CRS + 7,8-DHF? It would be important to test if intra-DRN of trkB agonist has similar antidepressant effect as seen with inhalation.

Reviewer #3 (Remarks to the Author):

This paper reports that treatment with mesenchymal stromal cells (MSCs) may alleviate depression/anxiety-like symptoms in mouse depression models, via lung vagal-to-brain axis. The authors treated immunocompetent B6 mice with human MSCs in two mouse depression models and observed activation of 5-HT neurons in the DRN and improved mouse behaviors. Further analyses indicated that MSCs may settle into the lung, co-localize near the lung vagal innervating nerves, and activate 5-HT neurons in the DRN via a lung-brain axis. Human dermal fibroblasts were used as control. The data are interesting and novel. However, there are several major concerns:

(1) The authors injected donor human MSCs intravenously into immunocompetent B6 mice. Human dermal fibroblasts were used as the control. Theoretically, immune cells in the immunocompetent mice will recognize and attack the human donor cells, triggering extensive host-versus-graft immune responses. Could the authors clarify why human MSCs can successfully engraft immunocompetent B6 mice without triggering extensive immune responses?

(2) Figure 3 demonstrated the presence of donor MSCs in the lung using fluorescence imaging. Could the authors verify whether these donor MSCs are still alive and functional in the mouse lung?

(3) In Figure 5a, RNA-seq was performed with human donor MSCs before administration into the mice. Could the authors clarify whether the donor MSCs still produce BDNF in vivo after settling in the mouse

lungs? In addition, the seq analyses is oversimplified, and lacks control.

(4) Figure 6 tests whether intranasal inhalation of a TrkB agonist is a promising depression treatment. Intranasal delivery may lead to direct entry into the CNS, bypassing the BBB. This should be taken into consideration by the authors.

Reviewer #4 (Remarks to the Author):

Huang et al propose that peripherally injected mesenchymal stromal cells (MSC) diminish anxiety and depressive behavior effects through a novel lung vagal-to-brain pathway. The proposed mechanism involves MSC-derived BDNF that triggers TrkB activation in pulmonary vagal sensory afferents, which in turn activate a serotonergic pathway from the NTS to the dorsal raphe nucleus (DRN). To this end, the authors demonstrated that MSC injection reduced depressive and anxiety-like behaviors in two conventional murine models of major depressive disorder (MDD): chronic restraint stress and repeated social defeat. They showed that the MSC therapeutic effect was diminished when 5-HTDRN neurons were ablated via 5,7-DHT or AAV. Further, they demonstrated that vagal innervation of the lung was required for central activation by peripheral MSCs with vagotomies. The functional pathway between lung MSCs and the brain was confirmed with viral tracing. They showed pulmonary BDNF expression was elevated in MSC-injected animals leading to increased TrkB expression in peripheral vagal neurons. This effect was ablated in siRNA BDNF downregulated MSCs. Finally, the authors reveal that inhalation of a TrkB agonist, 7,8-DHF, is sufficient to elicit the therapeutic effects seen in MSC-injected animals. Thus, the authors propose a novel therapeutic alternative to anxiety and depression medication by using MSCs as a pioneer. Overall, this study presents an intriguing approach using cell therapy as a framework for discovering novel therapeutics for difficult to treat disorders, such as MDD and anxiety.

Major Comments:

- Throughout the text the authors focus on MDD as the disorder they want to target for treatment, however the majority of the behavioral assessments throughout the manuscript (open field test and elevated plus maze) are used primarily to assess anxiety. Additional background on the need to treat anxiety would help bring the introduction in line with the data presented. Much of the results section acknowledges the relevance of both depressive and anxiety-linked behaviors, with the exception of the result header on lines 178 and 297 (and corresponding figure legend titles; Fig 2 and Fig 6).
- There is a lack of statistical rigor, or at least in the reporting of statistics in the manuscript. The statistical tests used should be reported in the Figure legends or results sections, and as many of the group distributions do not look normally distributed, it is not appropriate to use Student's t-test for all comparisons as stated in the Methods section.

Minor Comments:

- Figure 1b: control box is white but data points are black, recommended to use whisker box-plot to keep this figure in line with the style of the rest of the paper.

- Line 127: extra space after “mice .”
- Lines 131-133: the results state that RSD mice spent more time in and entry numbers to closed arms in the EPM test, however, the data in Supplementary data Fig 1D does not quantify entry numbers to closed arms, and Supplementary data Fig 1E refers only to time spent in open arms. Please rephrase this line to reflect the data presented.
- Figure 1j: The c-Fos relative protein expression y-axis starts at 0.5 instead of 0, recommended to put a line-break like in other graphs.
- Lines 161-163: Add a reference for this statement.
- Figure 3h: c-Fos control has a lot of background, which makes it difficult to compare to the c-Fos label neurons in the MSC
- Figure 3i: High background in sham group compared to Sham+MSC makes it difficult to compare groups visually.
- Lines 307-309: Would help to state that behavioral assessments were conducted on Day 15 as reflected on Fig. 6a.
- Lines 314-316: The phrasing of this sentence is unclear. It suggests that the 7,8-DHF treatment makes the treated mice more depressed, yet Fig. 6h, and Fig. 6i demonstrate the treated mice have lower immobile times.
- Line 343: The suggestion of “relatively sustained” antidepressant effect is only supported up to 6 hrs according to the Supplementary Fig. 8. A further kinetics study would strengthen this claim but generally this timescale would not be considered “sustained” in any real way.
- Lines 366-369: The link between the observed peripheral BDNF-TrkB signaling and CNS derived BDNF-TrkB antidepressant effects is tenuous. Some literature support would help strengthen this assertion. Please add the appropriate References to support this or delete this, as the BDNF result alone does not support this.
- Line 378: “Vagus nerves” should be rephrased to “Vagus nerve innervation of peripheral organs has...”.
- Line 661: replace “a Olympus” with “an Olympus”

RESPONSE TO REVIEWERS' COMMENTS

The four reviewers raised a number of constructive criticisms and suggestions. To fully address them, we performed additional experiments as well as implementing considerable changes to the manuscript. As a result, we believe the manuscript is much stronger. We wish to take this opportunity to thank the reviewers for their valuable input. Below, we summarize the reviewers' comments, and describe point-by-point how we have addressed them.

Reviewer 1

- *This manuscript by Huang et al. describes a previously uncharacterized mechanism by which intravenous infusion of mesenchymal stem cells attenuate stress and anxiety-associated behavior through pulmonary vagal nerve mediated serotonin circuit stimulation. Although vagus nerve stimulation is already an FDA-approved treatment for major depressive disorders (MDD), additional information regarding its' mechanism remains elusive. This study provides an intriguing potential point of interaction between pulmonary and neural interactions in the context of MSC infusion treatment of MDD. I find this manuscript clearly organized and well written, with appropriately controlled experiments that tested the key components of the hypothesis. However, I would like to see the follow points addressed before recommending this manuscript for publication at Nature Communications.*

Many thanks for these positive and encouraging comments. We are grateful for your thoughts and constructive comments which have helped clarify and improve the manuscript, and welcome any further suggestions you might have on the revised documents.

- *Major point 1: Despite the visualization of GFP+ cells in the lung, it is unclear whether the observed GFP+ cells retain MSC properties at the time of analysis.*

The authors should include immunohistochemical confirmation of MSC specific markers and the lack of markers of cell lineages (ie bone or adipocytes) that MSCs are capable of generating. This analysis is necessary to determine the source of secreted factors that contribute to the antidepressant effect.

Response: As suggested, we performed immunofluorescence staining of MSC specific markers in lung sections at 3 hours and 3 days after MSC transplantation in order to identify whether GFP+ cells retain MSC properties. We found GFP+ cells were positive for CD105 and CD90, while being negative for CD45 and CD19 (**Attached Figure 1a**). Moreover, we performed Alizarin red S staining and Oil Red O staining to detect the osteogenic and adipogenic fate of the transplanted GFP+ cells in mouse lungs and did not observe any positive signal, indicating that GFP+ cells did not differentiate into bone or adipocyte lineages in vivo (**Attached Figure 1b, c**). We also sorted the GFP+ MSCs using FACS at 3 days after the transplantation to compare the gene expression profiles to test whether the intrinsic characteristics of MSCs changed. The results showed they expressed the MSC related cell surface markers, like CD29, CD90 and CD105, and were negative for CD3, CD19, and CD34. Similar to the MSCs with conventional conditions in vitro, GFP-MSCs isolated from the lungs did not express the osteogenic, chondrogenic and adipogenic related genes. More importantly, GFP-MSCs from the lungs are still active as they express survival related genes and secreted protein genes. We added the corresponding data in the **Supplementary Fig. 8a-d**. Taken together, we speculate that the GFP+ MSCs in the lungs retain MSC properties at the time of analysis.

Attached Figure 1. GFP+ cells retain MSC properties.

a. Representative immunofluorescence images of the expression of MSC specific markers of GFP+ cells in lungs. Scale bar, 100 μm. b. Representative images of the Alizarin red S staining. Scale bar, 500 μm. c. Representative images of the Oil Red O staining. Scale bar, 500 μm.

- Major point 2:** *The authors previously published that MSC-induced cholinergic anti-inflammatory signaling could attenuate the effect of LPS induced lung inflammation. Although the authors did not observe changes in IL-6 and TNF-α levels after stressor, the increase in IL-6 after CRS or RSD is well-established in the literature. The absence of change in IL-6 here is likely due the difference in measuring at the acute (4 hrs) vs. chronic (11 days) time points after stressors. To properly assess the influence of MSC infusion on IL-6 and TNF-alpha levels, the experimental design needed to be adjusted to capture the increase in IL-6 post stress and determine whether the presence of MSC alters this response. Alternatively, downstream signaling (ie pSTAT3) or transcriptional targets (ie SOCS3) or physiological readout (ie reactive gliosis) of these signaling pathways should also be examined at the experimental end point to conclude that MSC infusion does not influence depressive behavior through effects on stressors-induced inflammatory responses.*

Response: This is a good point. As suggested, we detected IL-6 and TNF- α levels at the acute phase (4 hours) to properly assess the influence of MSC infusion on IL-6 and TNF- α levels. Unlike chronic phase (11 days) of CRS, acute phase induced a significant increase in IL-6 level (serum and brain homogenate supernatant), which could be alleviated by MSC injection (**Supplementary Fig. 2a, b**). In contrast, MSC injection barely affected the level of TNF- α at the acute phase (**Supplementary Fig. 2c, d**). To evaluate whether intravenous infusion of MSCs could modulate neuroinflammation in acute phase (4 hours) and chronic phase (11 days), we performed immunostaining to detect the activated microglia marker (Iba-1) in DRN. Our data showed that 4 hour stress significantly activated microglia, and MSC injection could alleviate neuroinflammation. However, we did not observe significant microglial activation during chronic phase (**Supplementary Fig. 2e, f**). Taken together, our data indicate that the inflammation level of mice depends on the stress duration, and only at the early stage of stress, MSCs showed a notable anti-inflammatory effect. Consistently, one elegant study demonstrated that infusion of MSCs on day 6 of stress demonstrated less effective modulation of both IL-6 and proinflammatory monocytes than the early delivery of MSCs on day 2 (*Biol Psychiatry*. 2019, 86:712-724). The corresponding data was added as **Supplementary Fig. 2**.

Accordingly, the majority of patients with chronic anxiety and depression are in an immunosuppressive state that is characterized by susceptibility to infection, and only a subset of patients with MDD show hyperactivation of innate immunity (*Biol Psychiatry*. 2020, 88:369-380). In addition, although inflammation has been considered as one of the important factors involved in depression progress, anti-inflammatory agents often show less significant or negative effects on depression score (*Lancet*. 2023, 401:141-153). Most recently, Ai M, et al demonstrated that MSC could improve osteoarthritis pain via direct modulation of sensory neurons innervating the joint, rather than their anti-inflammatory properties (*Arthritis Rheumatol*. 2023, 75:352-363). In this study, we identify the antidepressant role of peripheral MSC through the pulmonary vagal→

NTS→5-HT^{DRN} neural pathway in chronic stress-based models, which shed new light on developing novel treatments for major depressive disorder.

- *Major point 3: In addition to c-Fos staining, the electrophysiological recordings in panels 1o-1q provide additional support for MSC induced neuronal activation in DRN. However, controls with no MSC infusion and with fibroblast infusion should be included for the experiment to allow the interpretable results.*

Response: We performed the experiments as suggested. The firing rate of neurons in DRN was increased upon MSC infusion compared with control (no MSC fusion) and CRS group. Administration with HDFs could not increase the firing rate in the DRN. Patch-clamp recordings also suggested that MSC infusion could induce an increase in the firing frequency of 5-HT^{DRN} neurons. The corresponding data was added as **Supplementary Fig. 4a-c.**

- *Major point 4: I appreciate the authors used the two independent methods to ablate of 5-HT expressing neurons and find the results supporting that notion that 5-HT neurons are necessary for the MSC induced anti-depressant effect. However, the point about MSC inducing activation of 5-HT neurons can be more convincingly addressed with hM4Di silencing of 5-HT+ neurons in the presence of MSC infusion. Similarly, observation of 5-HT cell activation with virally delivered, hM3Dq-mediated vagus nerve stimulation would also strengthen this important point.*

Response: As suggested, we performed hM4Di silencing of 5-HT+ neuron experiments in order to identify whether MSC-induced activation of 5-HT neurons is necessary for the antidepressant effects of MSC to determine. We found MSC injection did not improve CRS-induced depressive and anxiety-like behaviors in mice with hM4Di silencing of 5-HT+ neurons. The corresponding data was added as **Supplementary Fig. 5b-h.**

Moreover, we performed hM3Dq-mediated vagus nerve stimulation and observed that

the stimulation of the vagus nerve could induce DRN 5-HT neuron activation (**Attached Figure 2**), which is consistent with the previous study that the serotonergic system was required for repeated vagal nerve stimulation (VNS)-induced antidepressant-like effects (*Biol Psychiatry*. 2011, 70:937-945).

Attached Figure 2. hM3Dq-mediated vagus nerve stimulation induced the activation of 5-HT^{DRN} neuron.

a. Schematic diagram showing the experimental procedures used for hM3Dq-mediated vagus nerve stimulation. Specifically, viral aliquots were loaded into a NanofilTM 36G beveled needle (WPI, Sarasota, FL) and SilflexTM tubing (WPI, Sarasota, FL). For each nodose ganglion, a total of 500 nL volume of virus was delivered at 50 nL/min using a NanofilTM 10 μ L syringe (WPI, Sarasota, FL) mounted on a UltraMicroPump3 (UMP3) pump head with SMARTouch Controller (WPI, Sarasota, FL). **b.** Representative immunofluorescence images showing the colocalization between c-Fos and 5-HT in the DRN after the chemogenetic activation. Scale bar, 100 μ m. **c.** The quantification of (b). Two-tailed t-test: AAV (n = 3) vs AAV+CNO (n = 3) ** $P = 0.0016$.

- *Major point 5: 7, 8-DHF crosses the blood-brain barrier and intranasal delivery of the agonist likely allow direct uptake of the drug through the olfactory epithelium/neurons into the brain through retrograde transport, thus the experimental approach does not limit effect of the drug specifically to vagal nerve endings in the lung. To support the hypothesis, the authors need to provide data that demonstrate the agonist's effects are through TrkB activation in the lung and not directly in the brain.*

Response: This is a good point. To verify 7, 8-DHF play the antidepressant role primarily through lung vagal-to-brain axis, we first deliver 7, 8-DHF through intratracheal infusion, which avoid the direct uptake through the olfactory epithelium/neurons into the brain. As shown in the **Supplementary Fig. 11 a-e**, the intratracheal infusion of 7, 8-DHF could improve the depressive and anxiety-like behaviors in the CRS mice. Moreover, we performed the unilateral vagotomy before intratracheal drug delivery, and found that the antidepressant effect of endotracheal infusion of 7, 8-DHF was significantly decreased. In addition, we found direct injection of 7, 8-DHF into DRN did not show significant antidepressant effects. Taken together, we conclude that the inhalation of TrkB agonist 7, 8-DHF plays the antidepressant role mainly by means of the vagal lung-to-brain axis. The corresponding data was added as **Supplementary Fig. 11**.

- *Minor point 1: Although briefly described in the Methods section, specific statistical analyses used for each experiment should be included in the figure legends.*

Response: As suggested, we have added specific statistical analyses in all the **Figure legends**.

- *Minor point 2: Increased hippocampal neurogenesis is frequently observed with anti-depressant action and is known to be promoted by elevated BDNF levels. It*

would be interesting for the authors to determine whether improved affective behavior is associated with changes in hippocampal neurogenesis.

Response: As suggested, we performed immunofluorescence staining of PCNA and DCX in order to identify whether intravenous administration of MSCs could promote hippocampal neurogenesis. However, we did not observe a notable increase in the expression of PCNA and DCX after MSC injection, indicating that the antidepressant effect of MSC might not be associated with the hippocampal neurogenesis (**Attached Figure 3**).

Attached Figure 3. The expression of PCNA and DCX in the hippocampus.

a. Representative immunofluorescence images showing the expression of DCX and PCNA in the hippocampus. Scale bar, 50 μm. Doublecortin, DCX; Proliferating Cell Nuclear Antigen, PCNA. **b.** The quantification of the expression of DCX. Two-tailed t-test: CRS (n = 3) vs CRS+MSC (n = 3) ns $P = 0.6599$. **c.** The quantification of the expression of PCNA. Two-tailed t-test: CRS (n = 3) vs CRS+MSC (n = 3) ns $P = 0.6483$.

- *Minor point 3: From a clinical application perspective, the safety and efficacy of intravenous mesenchymal infusion remain not fully understood. With the availability of VNS for MDD, it is difficult to imagine a scenario which MSC infusion pose significant advantage over VNS. While intranasal delivery of TrkB agonist is proposed as an alternative treatment strategy, discussion on the advantages and limitations of the approach in comparison to VNS should be included.*

Response: As suggested, we added one paragraph in **Discussion** section where we compared the advantages and limitations of different approaches (**Line 421-428**).

Reviewer 2

- *In this study, Huang and colleagues investigated the antidepressant effect of mesenchymal stromal cells (MSCs) in chronic restrain stress (CRS) mouse model of depression and anxiety. The authors showed that intravenous injection of MSCs displays significant antidepressant effects, evidenced by reversal efficacy of CRS-induced deficits in open field test (OFT), elevated plus maze (EPM) and tail suspension test (TST). They further showed that iv MSCs exert their effect through lung-vagal sensory neurons-NTS-DRN 5-HT neurons. At molecular level, they provide evidence supporting that BDNF-TrkB signaling plays important role in mediating MSCs-induced antidepressant effect. Overall, this is a very interesting study. Experiments are well executed and analyzed. However, there are some concerns that need to be thoroughly addressed.*

We are very grateful for your constructive and helpful comments and are glad that you think our research is very interesting.

- *Major point 1: Abbreviation HDF in Figure 1a should be explained in figure legend (human dermal fibroblast).*

Response: As suggested, we added more detailed description of human dermal fibroblasts (HDFs) in the **Figure legends**.

- *Major point 2: In Figure 1b, it is confusing what the top horizontal line indicates. One line should indicate the comparison between Control and CRS, and another one for comparison between CRS and MSC? Additionally, the labeling for groups is also confusing. My understanding is that Control, CRS, CRS+HDF, and CRS+MSC. If this is the case, the authors should provide injection groups of HDF alone and MSC alone in control mice (stress-naïve mice).*

Response: We apologize for the unclear descriptions. Actually, the labeling groups are Control, CRS, CRS+HDF, and CRS+MSC. As suggested, we revised the **Figure 1b** to make this point more clear. As suggested, we performed additional experiments to compare the changes in the MSC or HDF injection groups of control mice (stress-naïve mice) and found MSC or HDF did not change the weight and the time spent in the open arms in EPM test significantly. The corresponding data was added as **Supplementary Fig. 1f, g**.

- *Major point 3: The in vivo recording data are interesting. However, it is not clear what kind of mice they used to record DRN neurons: stress-naïve control or CRS-treated mice? It would be important to perform recording from both control and CRS neurons. Additionally, it is also unclear how the DRN 5-HT neurons were identified in those in vivo recordings, and what time after MSC treatment the recordings were performed. An interesting question is how long time the effect on DRN neurons lasts.*

Response: This is a good point. As you mentioned, we used CRS-treated mice to record DRN neurons as control to compare the differences between two groups (CRS and CRS+MSC) in the **Figure 1m**. As suggested, we also performed additional experiments to compare naïve control, CRS, CRS+HDF, and CRS+MSC groups, and the corresponding data was added as **Supplementary Fig. 4a-c**.

For the reviewer's second concern, according to the published articles (Science. 2022, 378:390-398, Pain. 2019, 160:136-150; Pharmacol Res. 2016, 113:81-91), 5-HT neurons exhibit a slow (0.1-4 Hz) and regular firing rate (coefficient of variation [COV], ranges from 0.12 to 0.87) and a broad biphasic (positive-negative) or triphasic waveform (0.8-3.5 ms; 1.4 ms first positive and negative deflections). We have added the image of a representative waveform in **Supplementary Fig. 4a**. In our experiment, we performed the in vivo recordings after 1 hour of MSC injection, and data were recorded for 30 minutes. We have added this information in the **Method** section.

For the reviewer's third concern, as shown in the **Supplementary Fig. 3g-h**, we performed a time course study for investigating c-Fos expression in the DRN for 0.5 - 24 hours following MSC injection. The expression of c-Fos showed a significant increase around 1.5 hours and reached the peak after 3 hours of MSC injection. After 6 hours, c-Fos expression still higher in MSC injection group than that of the control group, indicating that the effects of MSC on DRN neuron activation might be relatively sustainable.

- **Major point 4: In general, n for some experiments is low (Figure 1n, 1q, 1r). For Figure 1r, what is MSC group: MSC infusion alone or CRS + MSC infusion? It would be important for the authors to provide CRS + MSC infusion group. One interesting question is what the possible mechanism for the significant decrease of 5-HT in the CRS group of Figure 1r is. It would be useful to explore the intrinsic excitability of DRN 5-HT neurons in control, CRS, and CRS + MSC infusion groups.**

Response: Thank you for the suggestion. We added the sample numbers of **Figure 1n, q, r**. In **Figure 1r**. The MSC group means CRS+MSC infusion group. We made the correction in the revised manuscript.

For the reviewer's second concern, Sun N *et al* found, under depressive status, somatodendritic 5-HT_{1A}R_{auto}S in the DRN are hyperactive, causing reduced firing frequency of 5-HT neurons and low 5-HT level in the synaptic cleft (Science. 2022,

378:390-398), which might explain the significant decrease in the brain 5-HT level of **Figure 1r**.

As suggested, we performed the patch-clamp recording to identify the intrinsic excitability of DRN 5-HT neurons in control, CRS, and CRS+ MSC groups. Our data showed that CRS induced a decrease in the firing frequency of 5-HT neurons, and MSC infusion could enhance the firing of DRN serotonergic neurons (**Supplementary Fig. 4d-f**). We added the results in the manuscript to make this point more clear.

- *Major point 5: The results in Figure 1q showed that the fiber photometry recordings were obtained from DRN 5-HT neurons of GCaMP6-loxp mice. Validation data would be needed to show GCaMP6 expression selectively in 5-HT neurons.*

Response: This is a good point. We have performed immunostaining to confirm GCaMP6 expression selectively in 5-HT neurons. The corresponding data was added as **Supplementary Fig. 4g**.

- *Major point 6: It is unclear what is 7, 8-DHF group in Figure 6. Is it 7, 8-DHF alone or CRS + 7, 8-DHF? It would be important to test if intra-DRN of TrkB agonist has similar antidepressant effect as seen with inhalation.*

Response: We apologize for the unclear description. The 7, 8-DHF group means CRS+7, 8-DHF, and we have corrected all the unclear labeling in the revised manuscript. As suggested, we also performed additional experiments to identify whether intra-DRN injection of TrkB agonist has antidepressant effects in CRS mice. Surprisingly, we did not observe a significant antidepressant effect of intra-DRN TrkB agonist injection based on the EPM and TST. In detail, CRS mice with intra-DRN TrkB agonist injection showed a similar activity trajectory to the CRS mice with intra-DRN vehicle injection in the EPM. Moreover, intra-DRN TrkB agonist injection barely improved behavioral despair of CRS mice, as measured by the immobile time of TST. Consistently, we did not observe a notable increase in the brain 5-HT level of CRS mice after the intra-DRN

TrkB agonist injection. The corresponding data was added as **Supplementary Fig. 11f-j**. Taken together, we conclude that the inhalation of TrkB agonist 7, 8-DHF plays the antidepressant role mainly by means of the vagal lung-to-brain axis.

Reviewer 3

- *This paper reports that treatment with mesenchymal stromal cells (MSCs) may alleviate depression/anxiety-like symptoms in mouse depression models, via lung vagal-to-brain axis. The authors treated immunocompetent B6 mice with human MSCs in two mouse depression models and observed activation of 5-HT neurons in the DRN and improved mouse behaviors. Further analyses indicated that MSCs may settle into the lung, co-localize near the lung vagal innervating nerves, and activate 5-HT neurons in the DRN via a lung-brain axis. Human dermal fibroblasts were used as control. The data are interesting and novel. However, there are several major concerns:*

Thank you very much for your careful review and constructive suggestions. We hope these changes address the reviewer's comments sufficiently.

- *Major point 1: The authors injected donor human MSCs intravenously into immunocompetent B6 mice. Human dermal fibroblasts were used as the control. Theoretically, immune cells in the immunocompetent mice will recognize and attack the human donor cells, triggering extensive host-versus-graft immune responses. Could the authors clarify why human MSCs can successfully engraft immunocompetent B6 mice without triggering extensive immune responses?*

Response: This is a good point. One of the established tenets of immunity is that cells lacking the “self” markers, such as major histocompatibility complex (MHC) class I molecules, are quickly destroyed by immune responses. Therefore, there seems to be little to be gained from experiments in which human cells were infused into

immunocompetent mice. Surprisingly, an exception to this conclusion has come from experiments with human mesenchymal stromal cells (hMSCs). A large series of reports have demonstrated that hMSCs can effectively evade immune responses in immunocompetent mice (*Mol Ther.* 2017, 25:1748-1756). Because the culture-expanded hMSCs typically express low levels of MHC class I, and no MHC class II or co-stimulatory molecules (e.g., B7-1, B7-2, or CD40), a major consequence is that the xenogeneic mouse models can be used to assay the efficacy of hMSCs and thereby provide some preclinical data that are essential for well-designed trials in patients (*Nat. Biotechnol.* 2014, 32:252-260. *Cell Mol Immunol.* 2023, 20:555-557. *JCI Insight.* 2023, 8:e167402.). We fully agree with the reviewer's comment that control experiments are obviously critical in interpreting data and identifying the specific effects of hMSCs. As you mentioned, we used the human dermal fibroblasts (HDFs) as a control for in vivo study. As shown in the manuscript, there was no significant difference in the behavioral assessments and the 5-HT^{DRN} neuron activation between CRS mice and CRS+HDF mice. These data illustrate the specific therapeutic effects of MSCs on depression and anxiety.

- ***Major point 2: Figure 3 demonstrated the presence of donor MSCs in the lung using fluorescence imaging. Could the authors verify whether these donor MSCs are still alive and functional in the mouse lung?***

Response: This is a good point. We sorted the GFP-positive MSCs from mouse lungs using FACS at 3 days after the transplantation (**Attached Figure 4a**) and detected the cell viability using LIVE/DEAD™ (Invitrogen). We found that more than half of these cells were still alive (**Attached Figure 4b, c**). Consistent with the result, RNA-seq also indicated the GFP-MSCs from the lungs are still active as they still express essential viability genes (*Nat Rev Genet.* 2018,19:51-62), similar to GFP-MSCs in vitro (**Supplementary Fig. 8c**). Then, we performed immunofluorescence staining of BDNF in lung sections in order to identify the fate of transplanted MSCs. We found the colocalization of BDNF and the GFP signal, indicating the transplanted MSCs are still

alive and functional (**Supplementary Fig. 9a**). To further confirm this observation, using immunostaining and qPCR, we identified that the isolated GFP-MSCs from lungs could express BDNF at protein and mRNA level (**Supplementary Fig. 9b, c**). Accordingly, several studies also demonstrated the preferential migration and survival of intravascularly delivered MSCs in the lung (*Proc Natl Acad Sci U S A.* 2010,107:13724-9. *Stem Cells.* 2021,39:707-722. *Stem Cell Res Ther.* 2022,13:253). Taken together, we verify the transplanted MSCs are still alive and functional in mouse lungs.

Attached Figure 4. Donor MSCs are still alive and functional in the mouse lung.

a. Schematic diagram showing the experimental procedures to verify whether donor MSCs are still alive and functional in the mouse lung. **b.** The flow cytometry sorting were performed to isolate GFP+ cells from mice lungs after GFP-MSC injection. **c.** The flow cytometric analysis was performed to detect the cell viability of isolated GFP+ cells from mice lungs by using LIVE/DEAD™ (Invitrogen).

- **Major point 3: In Figure 5a, RNA-seq was performed with human donor MSCs before administration into the mice. Could the authors clarify whether the donor**

MSCs still produce BDNF in vivo after settling in the mouse lungs? In addition, the seq analyses is oversimplified, and lacks control.

Response: We fully agree with the reviewer's comments. Firstly, we performed immunofluorescence staining and identify the colocalization of injected GFP-MSCs and BDNF in mouse lung sections (**Supplementary Fig. 9a**). We then sorted the GFP-MSCs at 3 days after transplantation. Both the immunostaining and qRT-PCR proved these cells expressed BDNF (**Supplementary Fig. 9b, c**). The results suggest that the MSCs still could produce BDNF in vivo after administration.

For the reviewer's second concern, to elucidate MSCs have similar properties in vivo as in vitro, we performed the transcriptional profiles of MSCs cultured in the medium and isolated from lungs using bulk RNA sequencing (RNA-seq) analysis. We sorted the GFP+ MSCs using FACS at 3 days after the transplantation, the results showed they expressed the MSC specific markers, like CD29, CD90, and CD105, and were negative for CD3, CD19, and CD34. Similar to the MSCs with conventional conditions in vitro, GFP-MSCs isolated from the lungs did not express the genes of osteocytes, chondrocytes, and adipocytes. More importantly, GFP-MSCs from the lungs were still active according to expressing essential viability gene. MSCs isolated from lungs were still capable of secreting BDNF. We added the corresponding data in the **Supplementary Fig. 8 and Supplementary Fig. 9**. To discover the candidate targets participating in the MSC mediated lung afferent neuron activation, we analyzed the transcriptional profiles of MSCs. According to the Ingenuity Pathway Analysis (IPA) database, the venn diagram implicated that MSC-secreted protein, brain-derived neurotrophic factor (BDNF), was associated with major depression and anxiety disorders. Considering that BDNF is widely accepted for its involvement in resilience and antidepressant drug action (*Mol Psychiatry*. 2018,23:801-811.), we chose BDNF for the further study. The corresponding data was added in the **Figure 5a**.

- *Major point 4: Figure 6 tests whether intranasal inhalation of a TrkB agonist is a promising depression treatment. Intranasal delivery may lead to direct entry*

into the CNS, bypassing the BBB. This should be taken into consideration by the authors.

Response: This point echoes a similar point raised by reviewer #1. To verify 7, 8-DHF play the antidepressant role primarily through lung vagal-to-brain axis, we first deliver 7, 8-DHF through intratracheal infusion, which avoid the direct uptake through the olfactory epithelium/neurons into the brain. As shown in the **Supplementary Fig. 11 a-e**, the intratracheal infusion of 7, 8-DHF could improve the depressive and anxiety-like behaviors in the CRS mice. Moreover, we performed the unilateral vagotomy before intratracheal drug delivery, and found that the antidepressant effect of endotracheal infusion of 7, 8-DHF was significantly decreased. In addition, we found direct injection of 7, 8-DHF into DRN did not show significant antidepressant effects. Taken together, we conclude that the inhalation of TrkB agonist 7, 8-DHF plays the antidepressant role mainly by means of the vagal lung-to-brain axis. The corresponding data was added as **Supplementary Fig. 11**.

Reviewer 4

- *Huang et al propose that peripherally injected mesenchymal stromal cells (MSC) diminish anxiety and depressive behavior effects through a novel lung vagal-to-brain pathway. The proposed mechanism involves MSC-derived BDNF that triggers TrkB activation in pulmonary vagal sensory afferents, which in turn activate a serotonergic pathway from the NTS to the dorsal raphe nucleus (DRN). To this end, the authors demonstrated that MSC injection reduced depressive and anxiety-like behaviors in two conventional murine models of major depressive disorder (MDD): chronic restraint stress and repeated social defeat. They showed that the MSC therapeutic effect was diminished when 5-HTDRN neurons were ablated via 5,7-DHT or AAV. Further, they demonstrated that vagal innervation of the lung was required for central activation by peripheral*

MSCs with vagotomies. The functional pathway between lung MSCs and the brain was confirmed with viral tracing. They showed pulmonary BDNF expression was elevated in MSC-injected animals leading to increased TrkB expression in peripheral vagal neurons. This effect was ablated in siRNA BDNF downregulated MSCs. Finally, the authors reveal that inhalation of a TrkB agonist, 7,8-DHF, is sufficient to elicit the therapeutic effects seen in MSC-injected animals. Thus, the authors propose a novel therapeutic alternative to anxiety and depression medication by using MSCs as a pioneer. Overall, this study presents an intriguing approach using cell therapy as a framework for discovering novel therapeutics for difficult to treat disorders, such as MDD and anxiety.

Many thanks for this enjoyable description of the context of the paper!

- *Major point 1: Throughout the text the authors focus on MDD as the disorder they want to target for treatment, however the majority of the behavioral assessments throughout the manuscript (open field test and elevated plus maze) are used primarily to assess anxiety. Additional background on the need to treat anxiety would help bring the introduction in line with the data presented. Much of the results section acknowledges the relevance of both depressive and anxiety-linked behaviors, with the exception of the result header on lines 178 and 297 (and corresponding figure legend titles; Fig 2 and Fig 6).*

Response: We fully agree with the reviewer's comment. Depression often co-exists, at different levels of severity, with anxiety. Several epidemiological studies have suggested that depressive, and anxiety might be different presentations of a common latent phenomenon and might require common therapeutic approaches (Lancet. 2022,399:957-1022. Psychol Med. 2019,49:764-771.). As suggested, we have added background about the anxiety in the **Introduction** section and also revised the result headers to make the point more clear.

- *Major point 2: There is a lack of statistical rigor, or at least in the reporting of statistics in the manuscript. The statistical tests used should be reported in the Figure legends or results sections, and as many of the group distributions do not look normally distributed, it is not appropriate to use Student's t-test for all comparisons as stated in the Methods section.*

Response: We fully agree with reviewer's comment and apologize for the unclear descriptions for statistics in the **Method**. As suggested, we added the detailed description of statistical tests of each figure in the **Figure legends**. We also checked all the data showed in the picture, and chose the more appropriate method to perform statistical tests. Specifically, we checked the normality and homogeneity of variances before performing one-way ANOVA and t-test. When the normality assumption is violated, we performed the Kruskal-Wallis test (the nonparametric equivalent of the one-way ANOVA). Moreover, when the homogeneity of variances is violated, we performed the Brown-Forsythe ANOVA test.

- *Minor point 1: Figure 1b: control box is white but data points are black, recommended to use whisker box-plot to keep this figure in line with the style of the rest of the paper.*

Response: That's a good point. As suggested, we have used whisker box-plot in **Figure 1b**.

- *Minor point 2: Line 127: extra space after "mice".*

Response: We have made the correction and have also checked the manuscript carefully for additional spelling/grammatical errors.

- *Minor point 3: Lines 131-133: the results state that RSD mice spent more time in and entry numbers to closed arms in the EPM test, however, the data in Supplementary data Fig 1D does not quantify entry numbers to closed arms, and*

Supplementary data Fig 1E refers only to time spent in open arms. Please rephrase this line to reflect the data presented.

Response: We apologize for the unclear descriptions, and we have rephrased the performance of each group of mice in the EPM test (**Line135-136**).

- *Minor point 4: Figure 1j: The c-Fos relative protein expression y-axis starts at 0.5 instead of 0, recommended to put a line-break like in other graphs.*

Response: As suggested, we corrected the defect in **Figure 1j**.

- *Minor point 5: Lines 161-163: Add a reference for this statement.*

Response: As suggested, we have added a reference (Nat Commun. 2016 28;7:10503) for this statement (**Line 174**).

- *Minor point 6: Figure 3h: c-Fos control has a lot of background, which makes it difficult to compare to the c-Fos label neurons in the MSC.*

Response: We apologize for providing an atypical data. As suggested, we replaced them with the more representative figure of c-fos control in **Figure 3h**.

- *Minor point 7: Figure 3i: High background in sham group compared to Sham+MSC makes it difficult to compare groups visually.*

Response: As suggested, we replaced the images to make them comparable between groups in **Figure 3i**.

- *Minor point 8: Lines 307-309: Would help to state that behavioral assessments were conducted on Day 15 as reflected on Fig. 6a.*

Response: We performed behavioral assessments on Day 15 as reflected on **Fig. 6a**, as we observed a significant 5-HT level increase of mice brain the day after 7, 8-DHF

inhalation, as shown in the Fig. 6c. As suggested, we revised the sentence to make this point more clear (**Line 348**).

- *Minor point 9: Lines 314-316: The phrasing of this sentence is unclear. It suggests that the 7,8-DHF treatment makes the treated mice more depressed, yet Fig. 6h, and Fig. 6i demonstrate the treated mice have lower immobile times.*

Response: We apologize for our unclear description, and have revised the sentence (**Line 353-355**).

- *Minor point 10: Line 343: The suggestion of “relatively sustained” antidepressant effect is only supported up to 6 hrs according to the Supplementary Fig. 8. A further kinetics study would strengthen this claim but generally this timescale would not be considered “sustained” in any real way.*

Response: We fully agree with the reviewer’s comment. As shown in the **Supplementary Fig. 3g-h**, we performed a time course study for investigating c-Fos expression in the DRN for 0.5 - 24 hours following MSC injection. The expression of c-Fos showed a significant increase around 1.5 hours and reached the peak after 3 hours of MSC injection. After 6 hours, c-Fos expression still higher in MSC injection group than that of the control group, indicating that the effects of MSC on DRN neuron activation might be relatively sustainable. As suggested, we rephrased the sentence to make the point more clear (**Line 391-392**).

- *Minor point 11: Lines 366-369: The link between the observed peripheral BDNF-TrkB signaling and CNS derived BDNF-TrkB antidepressant effects is tenuous. Some literature support would help strengthen this assertion. Please add the appropriate References to support this or delete this, as the BDNF result alone does not support this.*

Response: We agree that the unclear description dilutes the focus. We therefore removed it from the paper.

- *Minor point 12: Line 378: “Vagus nerves” should be rephrased to “Vagus nerve innervation of peripheral organs has...”.*

Response: As suggested, we rephrased the sentence (**Line 429**).

- *Minor point 13: Line 661: replace “a Olympus” with “an Olympus”.*

Response: We have made the correction and have also checked the manuscript carefully for additional spelling or grammatical errors (**Line 752**).

REVIEWER COMMENTS

Reviewer #1 (Remarks to the Author):

In this revision of manuscript NCOMMS-23-05217A, the authors have added several new experiments and text clarification that strengthened the conclusions and improve the quality of the manuscript. Specifically, all my previously comments have been sufficiently addressed, and I have no additional objection in the acceptance and publication of this manuscript at Nature Communications. Here are a few minor comments, if addressed, can further improve the manuscript.

1. In supplemental figure 2e, the images provided does not allow visualization of iba1 positive cells in the unstressed condition, giving the impression that iba1 is not detected in DRN even though it is normally expressed by microglia at baseline levels. The images are suggestive of stress increasing iba1 levels and MSC prevented the stress-induced increase. It would be best to increase the sensitivity or exposure of the imaging across all conditions to accurately represent the Iba1-expressing cells at baseline.
2. The inclusion of hm4Di DREADD-mediated silencing of 5-HT neurons have further strengthened the conclusion that MSC-induced anti-depressant effects are mediated by 5-HT neurons in DRN. However, it is unclear whether the control group also received the AAV and saline (instead of CNO) injections, or were they stress-naïve mice without any manipulation. This information is important for interpretation of the results as it's possible that AAV/CNO injections induced observed stress response. The figure legend and experimental methods should be updated to include this information. The authors also did not include an experimental group that demonstrated MSC infusion remains capable of inducing antidepressant effects in AAV injected mice without CNO administration, making this experiment somewhat incomplete. However, I believe the data is a nice supportive piece of information and can be included without additional experiments.
3. Regarding the sustainability of MSC-induced anti-depressant effects, the authors address the question from reviewer 4 by examining the c-Fos expression in the DRN after MSC infusion. However, it's unclear how this change in c-Fos expression correlates with changes in affective behaviors associated with 7,8-HDF administration, as all behavioral tests were performed within 24hrs of treatments as far as I could discern. It will be interesting to explore in future studies how 7,8-HDF treatment compares to other rapid acting anti-depressants (ie Ketamine) regarding their behavioral improvement durations.
4. Line 281 "okanterogradly" should be "anterogradely".

Reviewer #2 (Remarks to the Author):

The authors are very responsive to the previous comments. They completely addressed the earlier concerns by including new experimental data and additional information. No further comments.

Reviewer #3 (Remarks to the Author):

The authors have appropriated addressed all my concerns.

Reviewer #4 (Remarks to the Author):

The authors have addressed all my comments and have appropriately revised the manuscript to strengthen their findings. I have no further comments or critiques.

RESPONSE TO REVIEWERS' COMMENTS

Reviewer 1

- *In this revision of manuscript NCOMMS-23-05217A, the authors have added several new experiments and text clarification that strengthened the conclusions and improve the quality of the manuscript. Specifically, all my previously comments have been sufficiently addressed, and I have no additional objection in the acceptance and publication of this manuscript at Nature Communications. Here are a few minor comments, if addressed, can further improve the manuscript.*

Response: Thanks for the reviewer's suggestions. According to your advice, we performed additional experiments as well as implementing corresponding changes to the manuscript. We hope that the revision will make the article more convincing.

- *In supplemental figure 2e, the images provided does not allow visualization of iba1 positive cells in the unstressed condition, giving the impression that iba1 is not detected in DRN even though it is normally expressed by microglia at baseline levels. The images are suggestive of stress increasing iba1 levels and MSC prevented the stress-induced increase. It would be best to increase the sensitivity or exposure of the imaging across all conditions to accurately represent the Iba1-expressing cells at baseline.*

Response: This is a good point. As you mentioned, iba1 is normally expressed by microglia in the brain regions of mice (Biol Psychiatry. 2019, 86:712-724). Consistently, we also observed the iba1 expression in the unstressed condition. In order to show the results more clearly, we increased the brightness of the images in the **supplementary Figure 2e**, which can be clearly seen from the figure that acute stress significantly activated microglia, as shown by the cell morphology and increased Iba-1 positive area in the DRN, while chronic stress did not show obvious microglial

activation. Importantly, our data indicate that the inflammation level of mice depends on the stress duration, and only at the early stage of stress, MSCs showed a notable anti-inflammatory effect.

- *The inclusion of hm4Di DREADD-mediated silencing of 5-HT neurons have further strengthened the conclusion that MSC-induced anti-depressant effects are mediated by 5-HT neurons in DRN. However, it is unclear whether the control group also received the AAV and saline (instead of CNO) injections, or were they stress-naïve mice without any manipulation. This information is important for interpretation of the results as it's possible that AAV/CNO injections induced observed stress response. The figure legend and experimental methods should be updated to include this information. The authors also did not include an experimental group that demonstrated MSC infusion remains capable of inducing antidepressant effects in AAV injected mice without CNO administration, making this experiment somewhat incomplete. However, I believe the data is a nice supportive piece of information and can be included without additional experiments.*

Response: Thanks for the reviewer's comments. As you mentioned, we used mice with AAV and saline without CNO injection as the control group, and we have added this information in the **Figure legends** and **Methods**. Our data showed that hm4Di DREADD-mediated silencing of 5-HT neurons, instead of the injection process, aggravated depressive and anxiety-like behaviors in mice. Moreover, as suggested, we also performed additional experiments to detect whether MSC infusion is capable of inducing antidepressant effects in AAV injected mice without CNO administration. Our results suggested that MSC injection showed a significant antidepressant effects in the AAV injected mice without CNO administration, which further strengthened our conclusion. The corresponding data was added as **Supplementary Fig. 5b-h**.

- *Regarding the sustainability of MSC-induced anti-depressant effects, the authors address the question from reviewer 4 by examining the c-Fos expression*

in the DRN after MSC infusion. However, it's unclear how this change in c-Fos expression correlates with changes in affective behaviors associated with 7,8-DHF administration, as all behavioral tests were performed within 24hrs of treatments as far as I could discern. It will be interesting to explore in future studies how 7,8-DHF treatment compares to other rapid acting anti-depressants (ie Ketamine) regarding their behavioral improvement durations.

Response: This is a good point. As suggested, we performed a time course study for investigating c-Fos expression in the DRN for 24 hours following the intratracheal infusion of 7, 8-DHF. Similar to the trend following MSC injection, the c-Fos expression reached the peak after 3 hours of intratracheal infusion of 7, 8-DHF, then gradually decreased. Interestingly, the c-Fos expression at 6 hours was still higher than that of the vehicle group, though there was no statistical difference. The corresponding data was added as **Supplementary Fig. 11a-b**.

We fully agree with the reviewer's comment about the future exploration on 7,8-DHF treatment comparing to other rapid acting anti-depressants (*ie*. Ketamine). Delayed onset is a serious limitation of classic antidepressants (Trends Cogn Sci. 2014, S1364-6613(14)00237-X). A low dose of ketamine was reported to produce antidepressant effects within hours for patients with refractory depression, and effectively reduce the suicidal tendency of patients (Am J Psychiatry. 2021, 178: 383-399). However, the potential addictive properties of ketamine and its risk for inducing schizophrenia have raised concerns (Pharmacol Rev. 2018, 70: 621-660). Therefore, scientists are still searching for new, fast-acting antidepressant targets and compounds. Notably, both typical and fast-acting antidepressants directly bind to TrkB, accounting for cell biological and behavioral actions of antidepressants (Cell. 2021, 184:1299-1313), which highlights the importance of TrkB signaling in antidepressants development. 7, 8-DHF, acting as a selective TrkB agonist, represents a novel bioactive therapeutic agent for treating neurological diseases and mental disorders (Curr Neuropharmacol. 2016, 14:721-731). Unlike ketamine, chronic treatment with 7,8-DHF has

demonstrated no detectable toxicity (*Neuropsychopharmacology*. 2012, 37:434-444), which makes 7, 8-DHF become a promising antidepressant drug. More interestingly, Chen *et al.* used a prodrug strategy for elevating 7,8-DHF oral bioavailability to promote the TrkB signaling-related drug development (*PNAS*. 2018, 115:578-583). Therefore, the future studies on the agents, promoting TrkB signaling activation, will inspire more good ideas on fast-acting and sustainable antidepressants. We agree that this is a good idea and have planned experiments along the suggested directions. Given the amount of time that these follow-up studies will likely require, we feel that they fall outside the current scope of this paper.

- *Line 281 “okanterogradly” should be “anterogradely”*

Response: We apologize for our oversight in preparing the manuscript, and have made the correction. We also checked the manuscript carefully for additional spelling/grammatical errors.

Reviewer 2

- *The authors are very responsive to the previous comments. They completely addressed the earlier concerns by including new experimental data and additional information. No further comments.*

Response: Thanks for the reviewer’s comments.

Reviewer 3

- *The authors have appropriated addressed all my concerns.*

Response: Thanks for the reviewer’s support to the study.

Reviewer 4

- *The authors have addressed all my comments and have appropriately revised the manuscript to strengthen their findings. I have no further comments or critiques.*

Response: Thanks for the reviewer's suggestions.

REVIEWERS' COMMENTS

Reviewer #1 (Remarks to the Author):

The authors have sufficiently addressed all of my concerns and I have not further comments. I recommend this manuscript for publication at Nature Communications.

REVIEWER COMMENTS

Reviewer #1 (Remarks to the Author):

The authors have sufficiently addressed all of my concerns and I have not further comments. I recommend this manuscript for publication at Nature Communications.

RESPONSE TO REVIEWERS' COMMENTS

Reviewer 1

- *The authors have sufficiently addressed all of my concerns and I have not further comments. I recommend this manuscript for publication at Nature Communications.*

Response: Thank you for the review's support to our manuscript.